# Stromal fibroblast growth factor 2 reduces the efficacy of bromodomain inhibitors in uveal melanoma

Vivian Chua[1,*] (iD), Marlana Orloff[2], Jessica LF Teh[1], Takahito Sugase[2], Connie Liao[1], Timothy J Purwin[1], Bao Q Lam[2], Mizue Terai[2], Grazia Ambrosini[3], Richard D Carvajal[3,4], Gary Schwartz[3,4], Takami Sato[2] & Andrew E Aplin[1,5,**] (iD)

## Abstract

Alterations in transcriptional programs promote tumor development and progression and are targetable by bromodomain and extraterminal (BET) protein inhibitors. However, in a multi-site clinical trial testing the novel BET inhibitor, PLX51107, in solid cancer patients, liver metastases of uveal melanoma (UM) patients progressed rapidly following treatment. Mechanisms of resistance to BET inhibitors in UM are unknown. We show that fibroblast growth factor 2 (FGF2) rescued UM cells from growth inhibition by BET inhibitors, and FGF2 effects were reversible by FGF receptor (FGFR) inhibitors. BET inhibitors also increased FGFR protein expression in UM cell lines and in patient tumor samples. Hepatic stellate cells (HSCs) secrete FGF2, and HSC-conditioned medium provided resistance of UM cells to BET inhibitors. PLX51107 was ineffective *in vivo*, but the combination of a FGFR inhibitor, AZD4547, and PLX51107 significantly suppressed the growth of xenograft UM tumors formed from subcutaneous inoculation of UM cells with HSCs and orthotopically in the liver. These results suggest that co-targeting of FGFR signaling is required to increase the responses of metastatic UM to BET inhibitors.

**Keywords** BET inhibitor; FGF2; FGFR; liver; uveal melanoma
**Subject Categories** Cancer; Chromatin, Epigenetics, Genomics & Functional Genomics

See also: **Benedetti & Altucci** (February 2019)

## Introduction

Targeted therapies for cancer inhibit pro-tumorigenic pathways that are frequently altered by somatic mutations. Although these therapies are largely efficacious, they are challenged by the development of resistance. An alternative therapeutic option is to target transcriptional dependencies which are associated with dysregulation of the expression/activity of chromatin regulators, transcription factors, and/or cofactors (Bradner *et al*, 2017). Inhibition of transcriptional dependencies is important as they are not typically identified by cancer genome sequencing and may be particularly relevant in cancers that have a low mutational burden (Bradner *et al*, 2017).

The BET family of proteins, BRD2, BRD3, BRD4 and BRDT, are chromatin readers that regulate transcription of genes by binding to acetylated lysine residues on tails of histones in chromatin (Filippakopoulos *et al*, 2012). BET proteins promote global transcriptional elongation (Winter *et al*, 2017). BRD4 is also able to localize to enhancers or super-enhancers and recruit transcriptional protein complexes including the positive transcription elongation factor, pTEFb, and Mediator for gene transcription (Jang *et al*, 2005; Filippakopoulos *et al*, 2010; Loven *et al*, 2013; Whyte *et al*, 2013). Additionally, BRD4 has been shown to bind to acetylated lysines of the transcription factors, NFκB and TWIST, and drive signaling-specific gene transcription (Brown *et al*, 2014; Shi *et al*, 2014). Inhibitors of BET proteins are potential anti-cancer agents and have been shown to suppress the growth of hematopoietic and solid tumors (Dawson *et al*, 2011; Lockwood *et al*, 2012; Ott *et al*, 2012; Segura *et al*, 2013; Ambrosini *et al*, 2015).

Uveal melanoma is the most common primary intraocular malignancy in adults in the United States. Approximately 50% of UM patients develop metastases, often with a delay of between 5 and 20 years following treatment of primary tumors. UM predominantly metastasizes to the liver (COMS, 2001; Rietschel *et al*, 2005;

1 Department of Cancer Biology, Thomas Jefferson University, Philadelphia, PA, USA
2 Department of Medical Oncology, Thomas Jefferson University, Philadelphia, PA, USA
3 The Herbert Irving Comprehensive Cancer Center, Columbia University Medical Center, New York, NY, USA
4 Division of Hematology/Oncology, Columbia University Medical Center, New York, NY, USA
5 Sidney Kimmel Cancer Center, Thomas Jefferson University, Philadelphia, PA, USA
*Corresponding author. Tel: +1 215 203 2857; E-mail: vivian.chua@jefferson.edu
**Corresponding author. Tel: +1 215 503 7296; E-mail: andrew.aplin@jefferson.edu

Ossowski & Aguirre-Ghiso, 2010). Currently, no therapies have been approved by the U.S. Food & Drug Administration (FDA) for metastatic UM highlighting an urgent unmet need. G-protein subunit alpha q/11 (*GNAQ/11*) mutations are identified in > 90% UM (Van Raamsdonk *et al*, 2009, 2010; Robertson *et al*, 2017) and induce constitutive activation of downstream signaling cascades such as the mitogen-activated protein kinase (MAPK), phosphoinositide 3-kinase/AKT (PI3K/AKT) and Yes-associated protein (YAP) pathways (Chua *et al*, 2017). These pathways could be therapeutic targets in metastatic UM; however, poor clinical responses to MAPK kinase (MEK) inhibitors have been reported (Carvajal *et al*, 2014, 2015).

As UM has a low mutational burden, this disease may be treated by inhibitors of transcriptional dependencies including BET inhibitors. Indeed, we have previously reported JQ1 to be highly effective in inhibiting the growth of *GNAQ/GNA11* mutant UM cells associated with downregulation of DNA damage response genes, *Bcl-xL* and *Rad51* (Ambrosini *et al*, 2015). However, JQ1 is not tested clinically due to its short half-life ($t_{1/2}$: 0.9–1.4 h) (Filippakopoulos *et al*, 2010) and resistance to other BET inhibitors has been reported despite pre-clinical success of BET inhibitors (Fong *et al*, 2015; Rathert *et al*, 2015; Kurimchak *et al*, 2016). A next-generation BET inhibitor, PLX51107 (Ozer *et al*, 2018; Plexxikon, Inc., Berkeley, CA), that has a half-life of 2.8 h and a broader therapeutic index, is currently being tested in clinical trials for patients with advanced malignancies including UM (NCT02683395). In UM, mechanisms of resistance to BET inhibitors are not known.

Resistance to therapies in UM may in part be due to effects of the secretome (e.g., growth factors, cytokines) from cells in the tumor microenvironment (TME). In this study, based on the lack of response of UM patients to BET inhibitor, we investigated regulation by growth factors in the TME on the efficacy of BET inhibitors. Using cell-based assays and patient samples, we demonstrate that paracrine secretion of FGF2 by stromal cells in the TME reduces the responses of UM to BET inhibition. Furthermore, we show evidence of elevated expression of receptors in the FGFR pathway in response to BET inhibition. Our findings *in vitro* and *in vivo* indicate that inhibition of the FGFR pathway improves the responses of metastatic UM to BET inhibitors.

# Results

### Advanced-stage UM patient tumor progression on PLX51107

A male patient in the PLX51107 clinical trial (patient #3) was diagnosed in April 2010 with choroidal melanoma in the left eye. He was treated with radioactive plaque, but in August 2013, the patient was confirmed to have developed metastases in the liver. The patient underwent intermittent immunoembolization between September 2013 and July 2015 (Fig 1A). He received pembrolizumab from February 2015 to August 2016. At around the same time, he was also given valproic acid until December 2016 and underwent chemoembolization from July 2016 to December 2016. The patient was then enrolled on the Phase 1b dose escalation study of the BET inhibitor, PLX51107, in September 2017 (Fig 1A). The patient received the drug for about a month until November 1, 2017, when significant progression of the disease in the liver was

observed (Fig 1B). A pre-treatment biopsy was collected from the liver metastases prior to the first cycle of PLX51107 treatment, and a post-treatment biopsy was obtained from the growing mass in the peritoneum shortly after removal of the patient from the protocol (Fig 1A).

### BET inhibitors reduce metastatic UM cell growth

We sought to determine mechanisms of resistance to BET inhibitor utilizing pre-clinical UM models. First, we characterized effects of PLX51107 on the viability of metastatic UM cell lines, UM001, UM004 and OMM1.3. A related BET inhibitor, PLX72853, and JQ1, which inhibits the growth of UM cell lines (Ambrosini *et al*, 2015), were also included in our studies. Following 8 days of treatment, BET inhibitors decreased UM001, UM004 and OMM1.3 cell colony growth in a dose-dependent manner (Fig 2A). Interestingly, PLX72853 was more potent than JQ1 and PLX51107 with inhibition of colony growth achieved at nanomolar concentrations (Fig 2A). From here onwards, according to the $IC_{50}$ of the BET inhibitors, we treated UM001 and OMM1.3 cells with 1 μM JQ1, 1 μM PLX51107 and 100 nM PLX72853, and UM004 cells with 2 μM JQ1, 2 μM PLX51107 and 200 nM PLX72853. To determine whether the decrease in colony growth induced by the BET inhibitors was associated with induction of apoptosis, we analyzed annexin V-APC levels. In comparison with DMSO controls, the BET inhibitors significantly increased apoptosis in UM001, UM004 and OMM1.3 cultures (Fig 2B).

### BET inhibitors alter expression of cell cycle regulators and apoptosis markers in UM

We further characterized effects of JQ1, PLX51107 and PLX72853 by performing reverse phase protein array (RPPA) which analyzes ~300 proteins and phospho-proteins (Fig EV1A). BET inhibitors downregulated the expression of proteins associated with cell cycle progression such as PLK1, cyclin B1, phospho-RB (S807/811), CDK1 and FOXM1 (Fig 2C). Other downregulated proteins included DUSP4, COX2 and HES1. These data were validated by Western blotting which confirmed downregulation of PLK1, cyclin B1 and phospho-RB (S807/811) following 48 h of BET inhibitor treatment (Fig 2D). We also observed downregulation of Wee1, which controls cell cycle progression and DNA damage repair, and SKP2, which promotes degradation of the cyclin-dependent kinase inhibitor p27 (Fig 2D). Conversely, in all cell lines treated with the BET inhibitors, p27 levels were upregulated, indicating that inhibition of BET proteins induces cell cycle arrest (Fig 2D). As the BET inhibitors promoted apoptosis of cells, we probed for the pro-apoptotic marker, cleaved PARP. Consistent with the annexin V-APC assay, BET inhibitors increased the expression of cleaved PARP (Fig 2D). Consistently, we observed downregulation of the expression of c-Myc and Rad51 following BET inhibitor treatment (Fig 2D).

### FGF2 provides resistance to BET inhibition in metastatic UM

Previously, we reported that hepatocyte growth factor (HGF) produced by stromal cells promoted resistance of metastatic UM cells to the MEK inhibitor trametinib (Cheng *et al*, 2015, 2017). Effects of the TME on the efficacy of epigenetic inhibitors are poorly

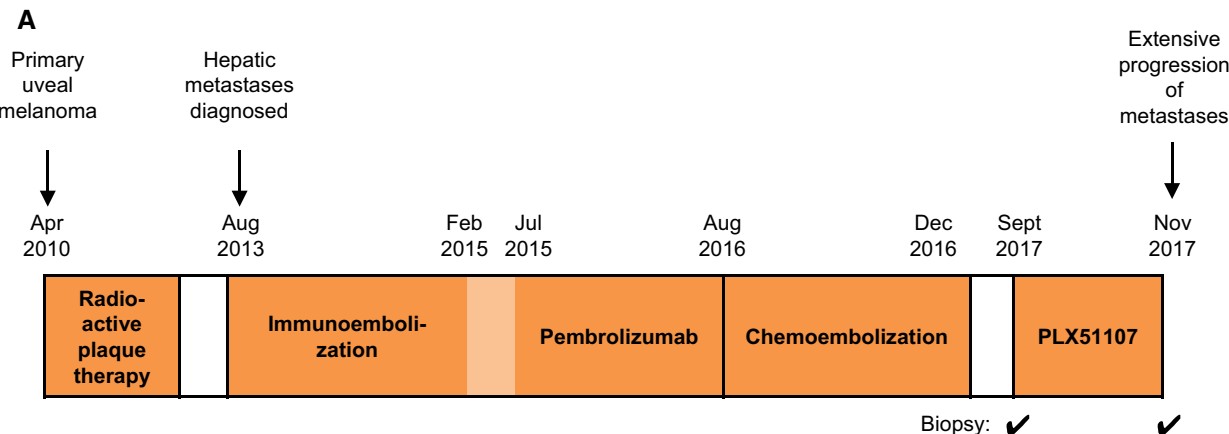

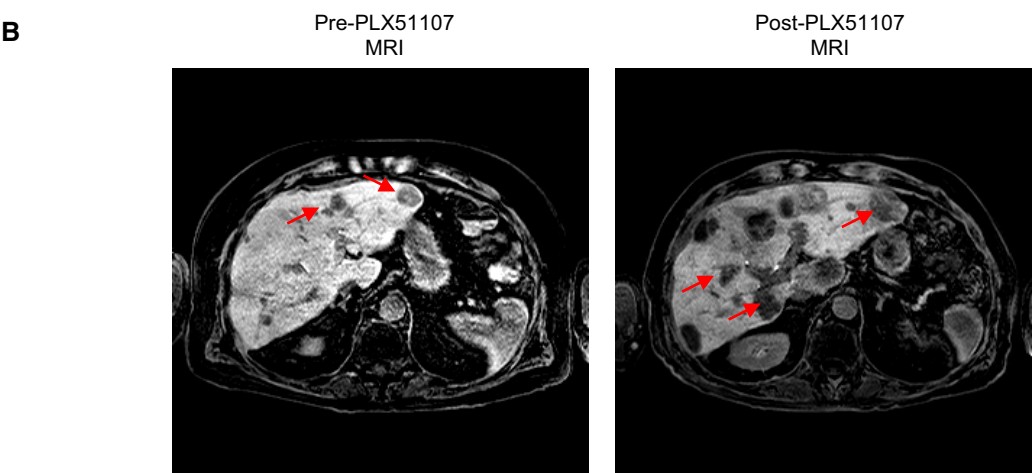

**Figure 1. PLX51107 clinical trial (patient #3).**

A  Treatment history of patient #3 in the PLX51107 trial. Biopsies were collected from metastases prior to treatment with PLX51107 and shortly after the patient was removed from the trial.

B  MRI scans of the patient's abdomen pre- and post-PLX51107 treatment. Increase in size and number of hepatic lesions (red arrows) were observed post-treatment.

characterized. We investigated a panel of growth factors that are known to be present in the liver microenvironment: FGF2, HGF, insulin-like growth factor 1 (IGF1) and neuregulin 1 (NRG1), and their effects on BET inhibitor response in metastatic UM (Fig 3A). BET inhibitor-induced suppression of UM001 viability was significantly rescued by FGF2, at low (1 ng/ml) to high (50 ng/ml) concentrations (Appendix Fig S1), whereas the other growth factors

had little effects (Fig 3A). In addition, vascular endothelial growth factor A (VEGF-A), fibroblast growth factor 1 (FGF1) and transforming growth factor alpha (TGFα) were tested. FGF1 rescued growth inhibition by BET inhibitors, but effects were weaker than FGF2 (Appendix Fig S2). In The Cancer Genome Atlas (TCGA), *GNAQ* Q209P and Q209L mutations are found in tumors of 32.5% and 12.5% UM patients, respectively. We verified FGF2 effects in a

**Figure 2. Inhibition of BET proteins in UM cells.**

A  UM001, UM004 and OMM1.3 cells were treated with increasing concentrations of JQ1, PLX51107 or PLX72853 for 8 days. Colony growth was examined by crystal violet staining. Representative images from at least three experiments are shown.

B  UM001 and OMM1.3 cells were treated with 1 μM JQ1, 1 μM PLX51107 or 100 nM PLX72853, and UM004 cells were treated with 2 μM JQ1, 2 μM PLX51107 or 200 nM PLX72853 for 48 h, and apoptotic cells were detected by annexin V-APC. The experiment was repeated three times, and mean ± SEM of data (*n* = 3) is shown. The unpaired *t*-test was used.

C  UM001 and OMM1.3 cells were treated with 1 μM JQ1, 1 μM PLX51107 or 100 nM PLX72853, and UM004 cells were treated with 2 μM JQ1, 2 μM PLX51107 or 200 nM PLX72853 for 48 h, and protein lysates collected for RPPA analysis. The RPPA data are log2 ratios of drug vs. DMSO for antibodies classified as significant in at least one cell line for JQ1 (*q*-value < 0.05 and fold ratio > 50%), PLX51107 and PLX72853.

D  UM001 and OMM1.3 cells were treated with 1 μM JQ1, 1 μM PLX51107 or 100 nM PLX72853, and UM004 cells were treated with 2 μM JQ1, 2 μM PLX51107 or 200 nM PLX72853 for 48 h, and then, lysates were analyzed by Western blotting. Blots shown are representatives from three replicate experiments.

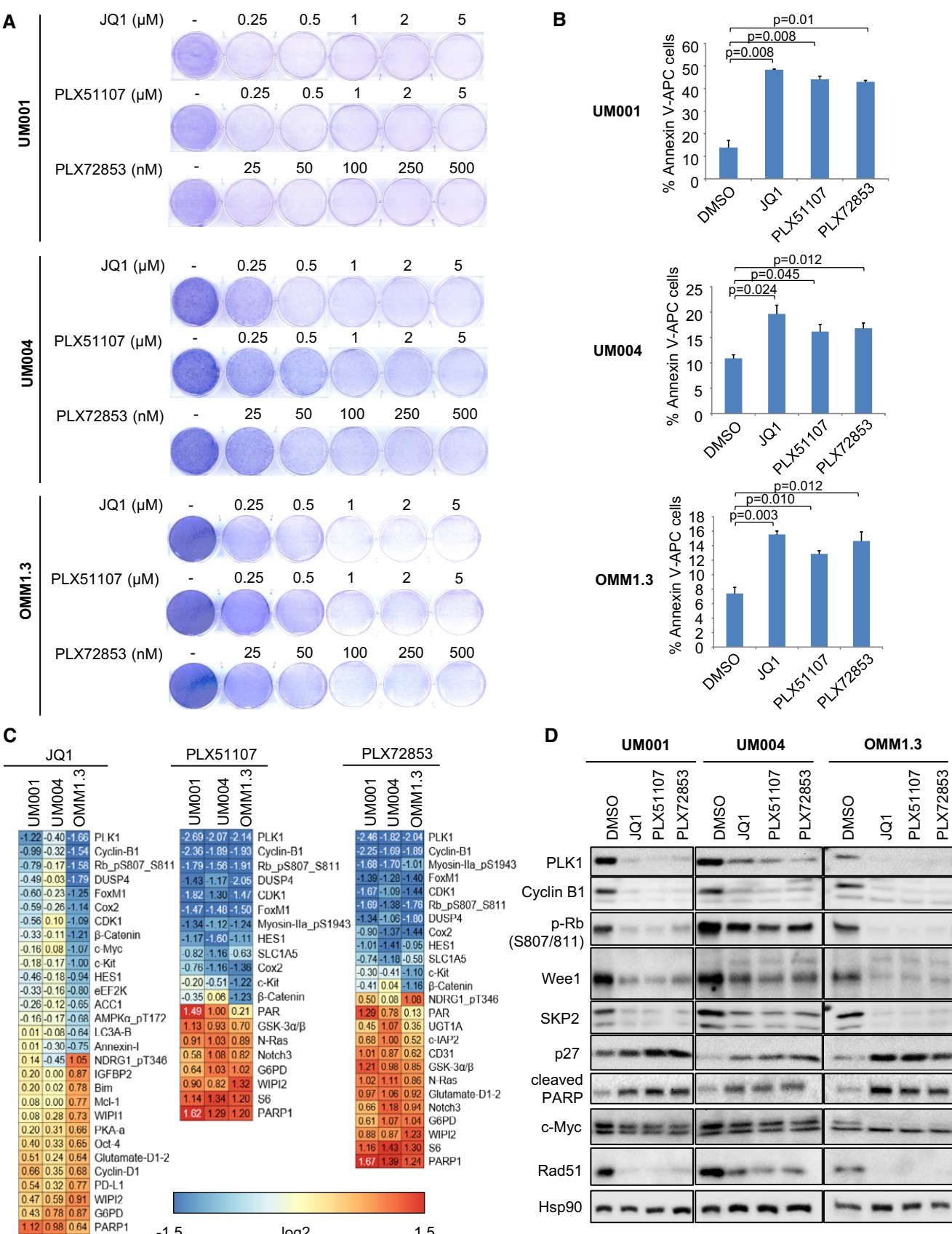

**Figure 2.**

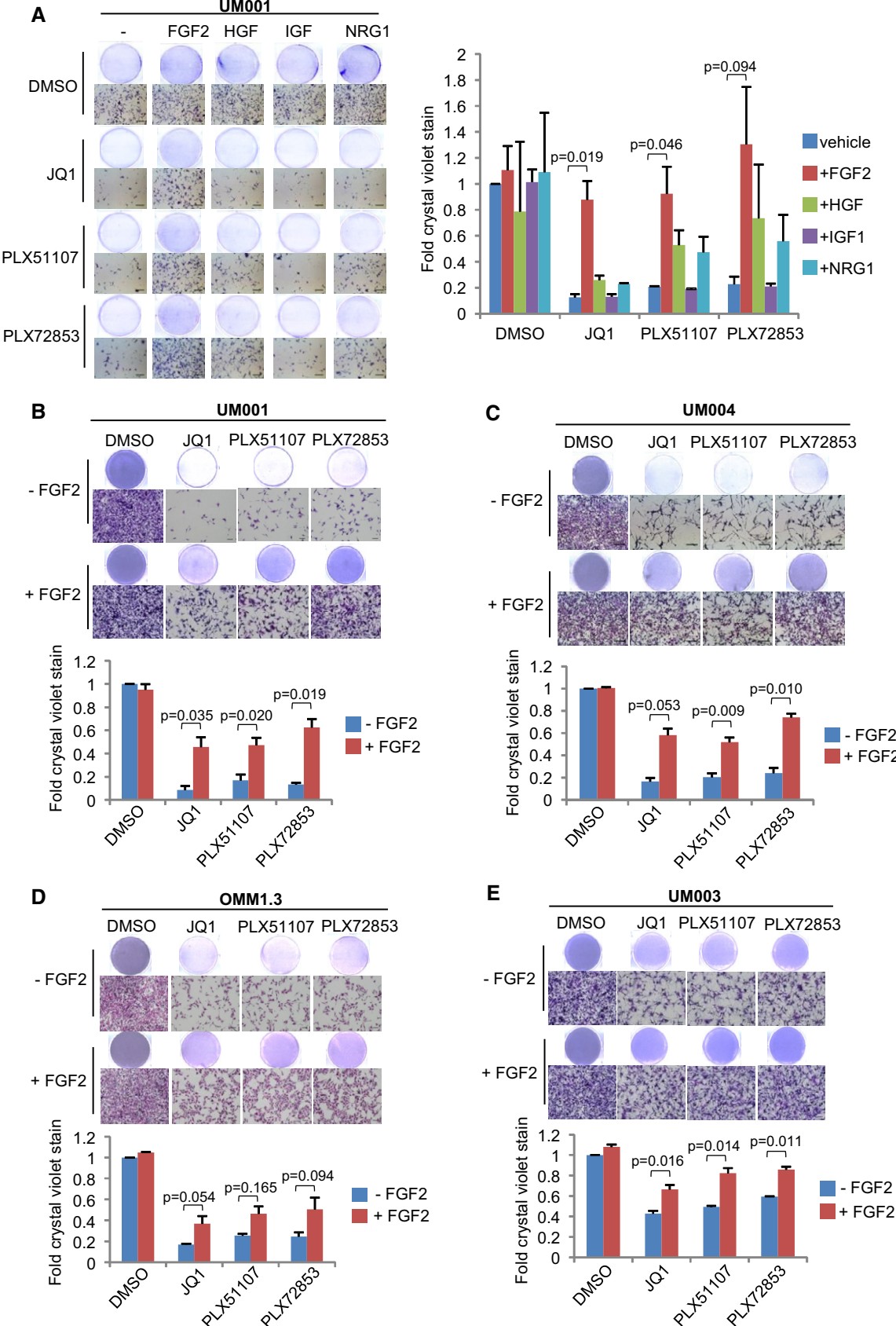

**Figure 3.**

**Figure 3.  FGF2 provides protection against BET inhibitor effects in UM cells.**

A–E   (A) UM001 cells were treated with 1 μM JQ1, 1 μM PLX51107 or 100 nM PLX72853 in combination with 50 ng/ml FGF2, 50 ng/ml HGF, 50 ng/ml IGF1 or 50 ng/ml
       NRG1 for 8 days. Changes in cell viability were determined by crystal violet staining. FGF2 rescued (B) UM001, (C) UM004, (D) OMM1.3 and (E) UM003 from the
       growth inhibitory effects of BET inhibitors. UM004 cells were treated with 2 μM JQ1, 2 μM PLX51107 or 200 nM PLX72853 in combination with 50 ng/ml FGF2 for
       8 days. Other cells were treated with 1 μM JQ1, 1 μM PLX51107 or 100 nM PLX72853 in combination with 50 ng/ml FGF2 for 8 days. Cell growth was determined
       by crystal violet staining. Data presented are fold change in crystal violet stain compared to untreated control and are mean ± SEM from triplicate experiments (or
       $n = 3$). The unpaired $t$-test was used for statistical significance. Representative crystal violet images are shown. Scale bar: 100 μm.

number of metastatic UM cell lines; UM004 (*GNAQ* Q209P), OMM1.3 (*GNAQ* Q209P) and UM003 (*GNAQ* Q209L). Consistently, in all cell lines, FGF2 provided protection against BET inhibitor effects on reducing colony growth (Fig 3B–E). FGF2-mediated rescue of BET inhibitor effects compared to BET inhibitor treatment alone was statistically significant in all lines except for OMM1.3 (*P*-value: 0.054–0.165; Fig 3B–E). These results indicate that FGF2 rescues metastatic UM cells from the growth inhibitory effects of BET inhibitors.

### FGF2 rescues BET inhibitor-induced apoptosis and cell cycle arrest

To investigate FGF2 effects on apoptosis and cell cycle arrest, annexin V and EdU incorporation assays were performed in UM001, UM004 and OMM1.3. FGF2 reversed BET inhibitor-mediated increase in percentage of annexin V-positive UM001, UM004 and OMM1.3 cells, indicating that FGF2 consistently reduces BET inhibitor-induced apoptosis (Fig 4A). Additionally, JQ1, PLX51107, and PLX72853 decreased the percentage of EdU incorporation, indicating inhibition of DNA synthesis/S-phase entry (Fig 4A). However, when cells were co-treated with the BET inhibitors and FGF2, EdU incorporation in UM001 and UM004 was significantly upregulated compared to samples treated with BET inhibitors alone while minimal effects on EdU incorporation were observed in OMM1.3 (Fig 4A). To study changes in the cell cycle, propidium iodide (PI) labeling of UM001 was determined (Appendix Fig S3). Interestingly, we did not identify marked changes in the percentage of cells in G0/G1 or S phases following BET inhibitor and/or FGF2 treatment but BET inhibitors increased the percentage of cells in sub-G1 which was reversed by FGF2. FGF2 also moderately rescued BET inhibitor-induced decrease in the percentage of cells in G2/M. These results indicate that FGF2 rescues cell cycle arrest in BET inhibitor-treated cell lines.

To understand signaling/protein expression changes associated with FGF2 regulation of apoptosis and cell cycle arrest in UM cells treated with BET inhibitors, we performed RPPA

(Fig EV1B). FGF2 reversed BET inhibitor-induced decrease of the expression of cell cycle proteins CDK1, cyclin B1, PLK1 and phospho-RB (S807/811) in all three cell lines, although effects were strongest in UM001 and UM004 (Fig 4B). We confirmed these findings by Western blotting (Fig 4C). We also showed that FGF2 reversed BET inhibitor-induced decrease of cyclin A2 and cyclin D1 expression (Fig 4C). FGF2 effects on protein expression were generally weaker in OMM1.3 cells, consistent with the RPPA data and findings from EdU incorporation assays (Fig 4A). BET inhibitors increased the expression of pro-apoptotic proteins; cleaved PARP, BimEL and Bmf, and these changes were also reversed by FGF2 treatment (Fig 4C). FGF2 also increased BET inhibitor-induced downregulation of expression of Bid, which has been reported to have anti-apoptotic effects (Luo *et al*, 2010; Appendix Fig S4).

### FGF2-induced resistance to BET inhibitors is mediated by FGFRs

To determine whether FGF2-induced resistance to BET inhibitors in the metastatic UM cell lines is mediated via FGFRs, we evaluated the FGFR1/2/3 inhibitor, AZD4547, and a FGFR4-specific inhibitor, BLU9931. FGF2 protects UM cells from the growth inhibitory effects of BET inhibitors as before and AZD4547 significantly suppressed FGF2-induced resistance of cells to BET inhibitors (Fig 5). Consistent results were observed in all three cell lines as well as in UM003 (Appendix Fig S5). BLU9931 had a moderate effect in reversing resistance to BET inhibition conferred by FGF2 (Fig EV2). AZD4547 or BLU9931 alone at 1 μM had little effect on cell growth (Fig EV3). These results indicate that the FGFRs, predominantly FGFR1/2/3, mediate FGF2-induced rescue of BET inhibitor effects in UM.

### HSC-conditioned media rescues BET inhibitor effects

Approximately 90% of UM metastases are diagnosed in the liver (COMS, 2001; Rietschel *et al*, 2005). HSCs are quiescent, lipid-storing cells located in the perisinusoidal space of the liver and undergo activation into myofibroblasts which are capable of supplying tumor cells in the liver with growth factors (Kang *et al*, 2011).

**Figure 4.  Effects of FGF2 on apoptosis and cell cycle progression in BET inhibitor-treated UM cells.**

UM001 and OMM1.3 cells were treated with 1 μM JQ1, 1 μM PLX51107 or 100 nM PLX72853, and 50 ng/ml FGF2 for 48 h. UM004 was treated with 2 μM JQ1, 2 μM
PLX51107 or 200 nM PLX72853, and 50 ng/ml FGF2 for 48 h.

A       Cells were collected for detection of apoptotic cells by annexin V-APC or for detection of DNA synthesis in the cell cycle by EdU incorporation. Mean of data from
        triplicate experiments (or $n = 3$) is shown. The unpaired $t$-test was used.
B, C   Lysates of (i) UM001, (ii) UM004 and (iii) OMM1.3 cells were collected for (B) RPPA analysis and (C) Western blotting. The RPPA data are median-centered log2-
        transformed group averages for antibodies classified as significant in at least one comparison (*P*-value < 0.05 and fold ratio > 50%). The Gene Ontology database
        was used to determine antibodies involved in regulation of cell cycle and cell differentiation. Representative Western blots from triplicate experiments are shown.

Source data are available online for this figure.

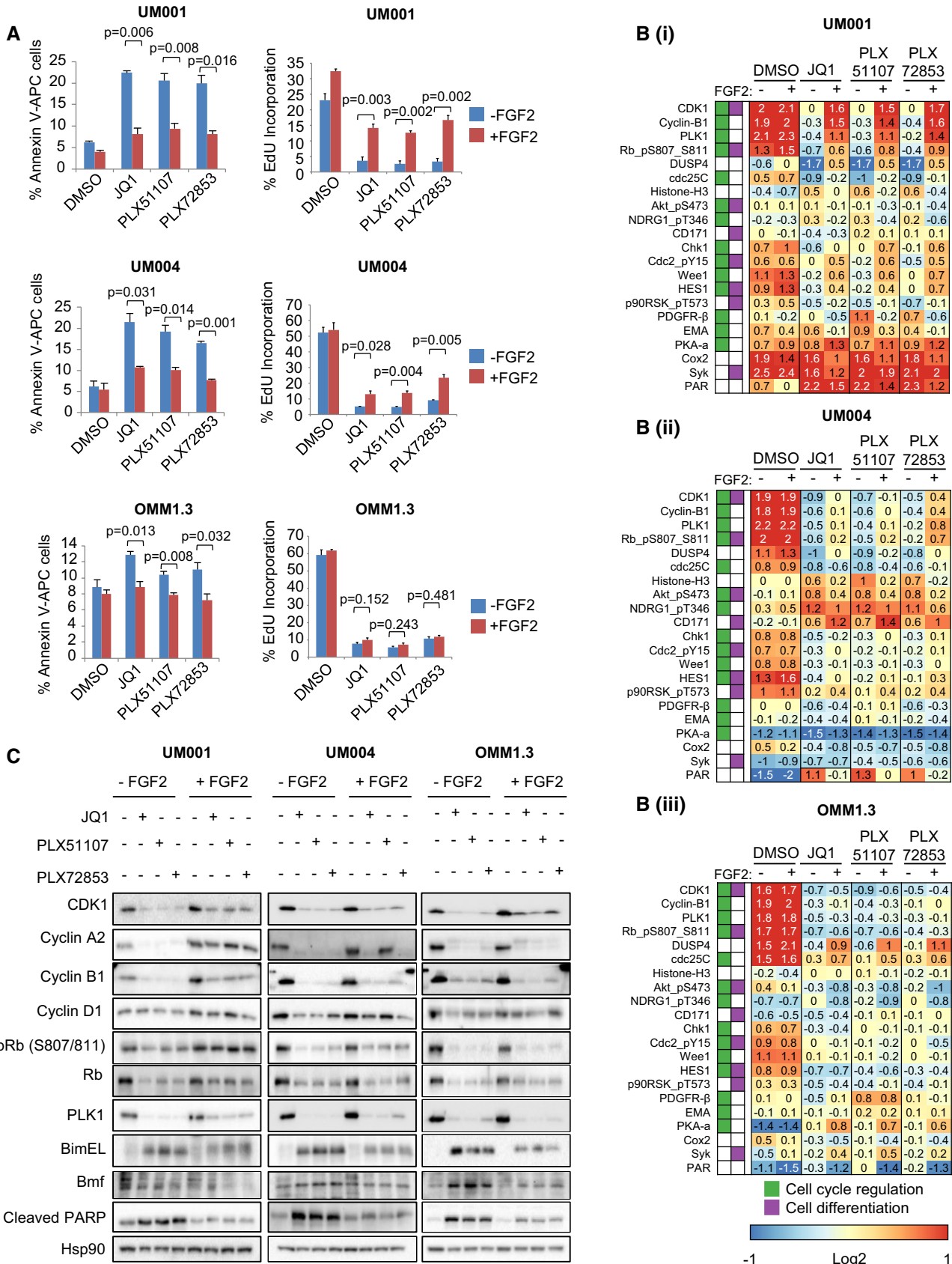

**Figure 4.**

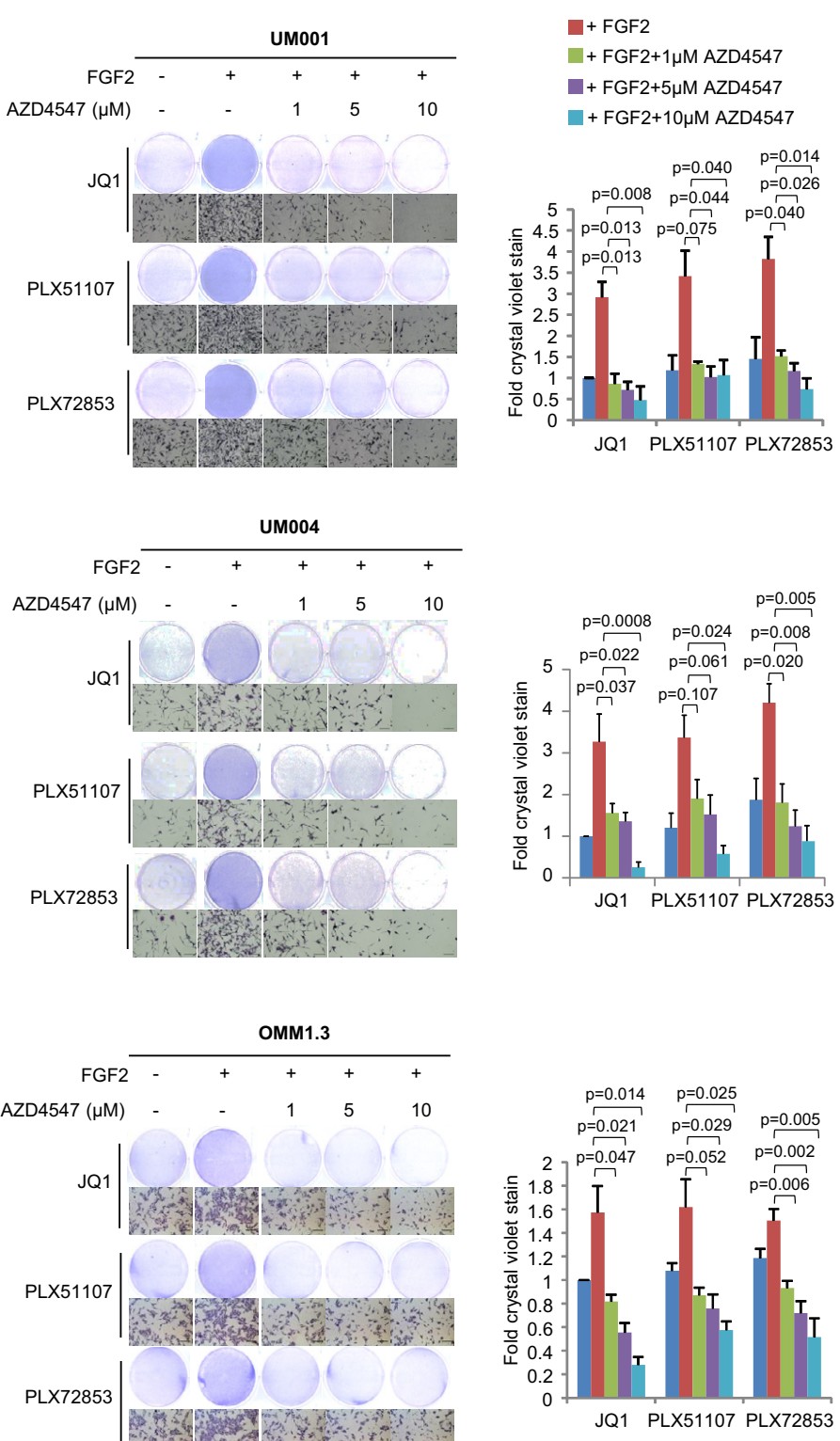

**Figure 5.   Inhibition of FGFR1/2/3 by AZD4547 reverses FGF2-induced rescue of BET inhibitor effects.**

UM001 and OMM1.3 cells were treated with 1 μM JQ1, 1 μM PLX51107 or 100 nM PLX72853 in combination with 50 ng/ml FGF2 and 0–10 μM FGFR1/2/3 inhibitor AZD4547. UM004 was treated with 2 μM JQ1, 2 μM PLX51107 or 200 nM PLX72853 in combination with 50 ng/ml FGF2 and 0–10 μM AZD4547. Cell growth was determined after 8 days of treatment by crystal violet staining. Fold change in crystal violet stain compared to JQ1 treatment and mean ± SEM of data from triplicate experiments (or $n = 3$) is shown. The unpaired *t*-test was used to calculate statistical significance. Scale bar: 100 μm.

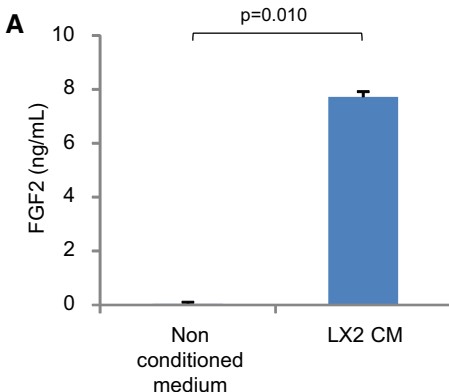

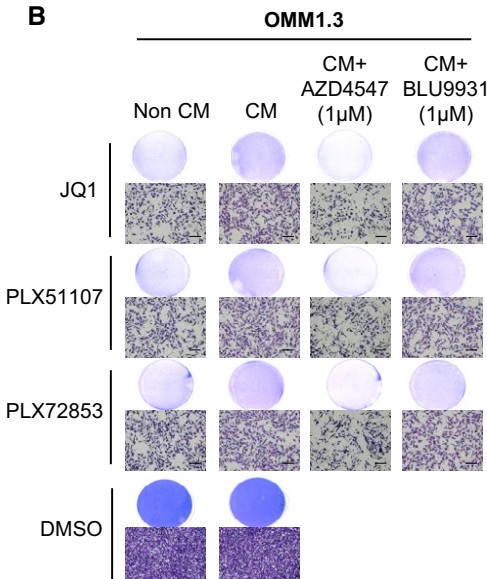

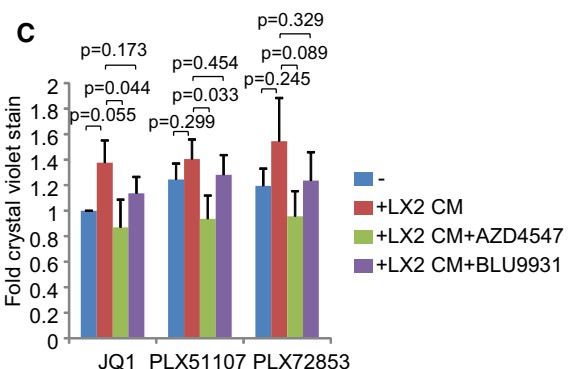

**Figure 6.  The LX-2 HSC line secretes FGF2.**

A  FGF2 in medium conditioned by LX-2 cells was measured by ELISA. Mean ± SEM from data (n = 3) is shown. The unpaired t-test was used.

B  OMM1.3 cells were treated with LX-2-conditioned medium in combination with 1 μM JQ1, 1 μM PLX51107 or 100 nM PLX72853, and 1 μM AZD4547 or 1 μM BLU9931 for 8 days. Cell growth was determined by crystal violet staining.

C  Data presented are fold change in crystal violet stain compared to non-conditioned medium/JQ1 treatment and are mean ± SEM from n = 4 experiments. The unpaired t-test was used. Representative crystal violet images are shown. Scale bar: 100 μm.

To determine whether FGF2 is produced by stromal cells in the liver, we measured FGF2 levels in media conditioned by the activated HSC line, LX-2 (Xu *et al*, 2005). In comparison with non-conditioned medium, there was a significant increase in FGF2 concentration in media conditioned by LX-2 cells by ELISA (Fig 6A). OMM1.3 cells were treated with the LX-2-conditioned media in combination with JQ1, PLX51107 or PLX72853 for 8 days. LX-2-conditioned media induced a moderate rescue of cells from BET inhibitor-mediated growth inhibition (Fig 6B and C). This effect was reversed by AZD4547, a pan-FGFR1/2/3 inhibitor, suggesting that the moderate resistance of OMM1.3 to BET inhibitors induced by the LX-2-conditioned media is mediated through FGFR1, 2, and/or 3.

### BET inhibitors increase production of FGF2 by HSCs and upregulate FGFR expression in metastatic UM

JQ1 has been reported to induce re-programming of receptor tyrosine kinases (RTKs) in ovarian cancer (Kurimchak *et al*, 2016). To investigate whether BET inhibitors alter FGF2/FGFR signaling, we determined effects of PLX51107, JQ1 and PLX72853 on FGF2 secretion by HSCs and FGFR expression in metastatic UM cell lines. Treatment of LX-2 cells with BET inhibitors increased the concentration of FGF2 in conditioned media from HSCs (Fig 7A). In UM cell lines, BET inhibitors upregulated FGFR protein expression, although heterogeneous effects were detected (Fig 7B). In UM001, BET inhibitor treatment increased expression of all FGFRs (FGFR1-4). By contrast, only FGFR2 was upregulated in UM004 and only FGFR1 was elevated in OMM1.3 (Fig 7B). These findings suggest that in addition to intrinsic resistance to BET inhibitors by FGF2 in the TME, BET inhibition induces adaptive response mechanisms that increase FGF2 production by HSCs and FGFR expression in UM cells. Furthermore, heterogeneity occurs in the specific FGFRs upregulated.

We also determined whether BET inhibitors alter FGFR expression in tumor samples using biopsy specimens from patients enrolled in the PLX51107 clinical trial. Pre- and post-PLX51107 treatment samples were analyzed by immunohistochemical staining for FGFR proteins. In patient #3 (Fig 1), we identified that in the post-treatment biopsy tissue, the intensity of FGFR1 staining increased compared to staining in the pre-treatment sample (Fig 7C). We also obtained pre- and post-treatment liver biopsies from a second patient (#2). Staining for FGFR1 and FGFR4 were negative, and FGFR2 staining was detected in both pre- and post-PLX51107-treated samples (Fig EV4).

### BET and FGFR inhibitors suppress *in vivo* UM tumor growth

Finally, we investigated effects of BET and FGFR inhibitors *in vivo* on UM001 xenografts formed from subcutaneous injection and in a liver orthotopic mouse model (Ozaki *et al*, 2016). Mice bearing UM001 xenograft tumors following subcutaneous injection of UM001 and LX-2 cells were treated with PLX51107 chow, AZD4547 or the combination of PLX51107 and AZD4547. LX-2 cells have been shown previously to not form into tumors *in vivo* (Amann *et al*, 2009; Barcena *et al*, 2015). Interestingly, PLX51107 increased UM001 tumor volume compared to controls (Fig 8A). AZD4547 moderately decreased UM tumor volume, but the combination of PLX51107 and AZD4547 significantly suppressed tumor growth

**A**

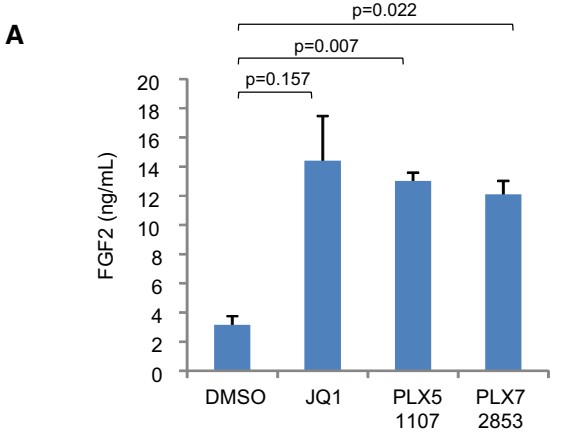

**Figure 7.    BET inhibitor effects on FGF2 secretion by LX-2 cells and FGFR expression in UM cells.**

A    LX-2 cells were treated with 1 μM JQ1, 1 μM PLX51107 or 100 nM PLX72853 for 48 h and FGF2 levels in LX-2-conditioned media measured by ELISA. FGF2 levels shown are mean of data from triplicate experiments (or n = 3). The unpaired t-test was used.

B    UM001 and OMM1.3 cells were treated with 1 μM JQ1, 1 μM PLX51107 or 100 nM PLX72853, and UM004 cells were treated with 2 μM JQ1, 2 μM PLX51107 or 200 nM PLX72853 for 48 h and then cell lysates collected for immunoblotting of FGFRs. Western blots shown are representatives from triplicate experiments.

C    Pre- and on-treatment tissue biopsies from patient #3 enrolled in the PLX51107 clinical trial (Fig 1) were fixed and stained for FGFRs. Arrows are indicating positive FGFR staining. FGFR1 and FGFR2 antibody concentrations were 1:75 and 1:50, respectively. Representative images are shown. Scale bar: 100 μm.

Source data are available online for this figure.

**B**

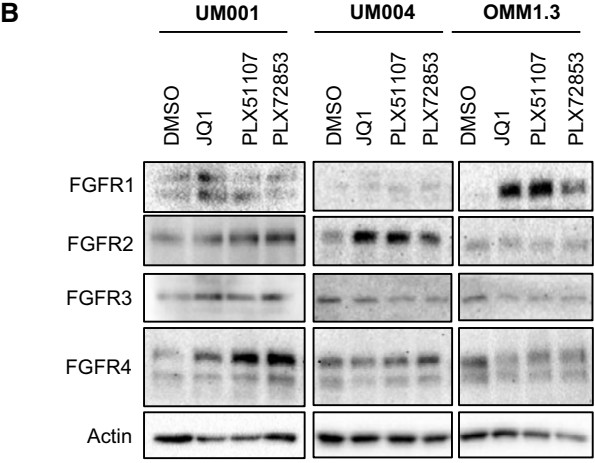

8 weeks, the animals were treated with PLX51107 chow, AZD4547 or the combination of PLX51107 and AZD4547 for a further 2 weeks. In comparison with the control, PLX51107 and AZD4547 alone suppressed tumor size moderately but the combination of PLX51107 and AZD4547 significantly decreased tumor size after 2 weeks of treatment (Fig 8B). These findings indicate that effects of BET and FGFR inhibitors as monotherapies are poor *in vivo* but the combination of both inhibitors suppressed UM tumor growth.

## Discussion

Inhibition of BET proteins is emerging as a promising anti-cancer therapeutic strategy to block transcriptional dependencies. The investigation of BET inhibitors has also been translated to clinical trials for treatment of advanced malignancies; however, the development of resistance remains a challenge. In both hematological and solid cancers, resistance to BET inhibition has been attributed to a number of mechanisms such as activation of pro-tumorigenic pathways, e.g., WNT/β-catenin signaling (Fong *et al*, 2015; Rathert *et al*, 2015; Kurimchak *et al*, 2016). Here, we studied potential mechanisms of resistance to BET inhibition in UM focusing on effects of growth factors in the TME. We found that FGF2, but not other growth factors, significantly provided resistance to growth suppression by BET inhibitors in metastatic UM cell lines. We observed promotion of apoptosis and cell cycle arrest in BET inhibitor-treated UM cells, but these effects were rescued strongly by FGF2. In addition, our findings indicated that stromal FGF2 rescues BET inhibitor effects in metastatic UM cells and also showed upregulation of FGFR expression as an adaptive response to BET inhibitors in cell lines and patient tumor samples. Previously, we reported that HGF secreted from HSCs rescued metastatic UM cells from the growth inhibitory effects of a MEK inhibitor, trametinib, and was associated with paracrine activation of cMET and PI3K/AKT pathways (Cheng *et al*, 2015, 2017). Our findings highlight the ability of stromal cells in the TME to mediate resistance to targeted inhibitors in UM.

We demonstrated that FGFR inhibitors, AZD4547 and BLU9931, reversed FGF2-induced protection against BET inhibitors, indicating that FGF2 effects are mediated by the canonical FGFRs. AZD4547, a pan-FGFR1/2/3 inhibitor, had a more significant effect on reversing FGF2-induced resistance to BET inhibition in the UM cell lines

**C**

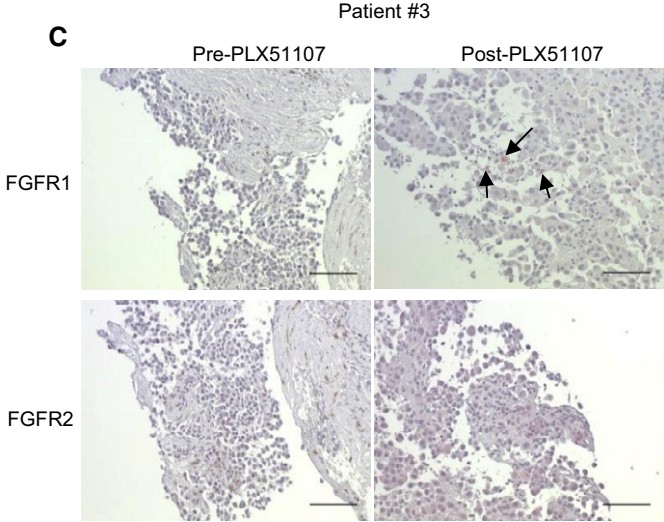

Patient #3

compared to the PLX51107 treatment arm (Fig 8A). Additionally, BimEL and cleaved PARP protein levels increased more rapidly in tumors from PLX51107 and AZD4547-treated mice compared to either PLX51107-treated or control mice (Appendix Fig S6). Next, we utilized a liver metastatic site model. UM001 cells were injected orthotopically to the liver of NSG mice (Ozaki *et al*, 2016). After 6–

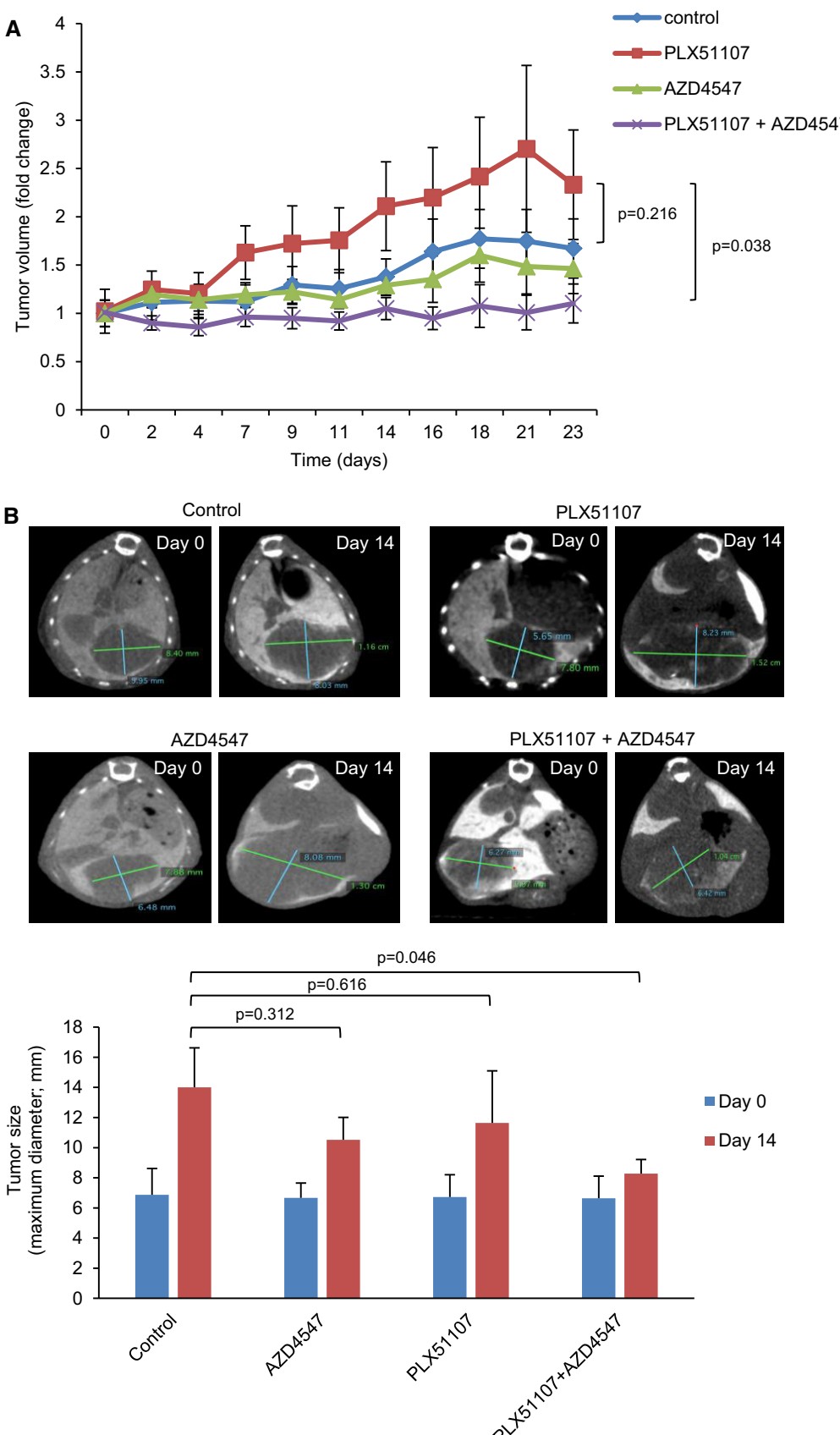

**Figure 8.**

**Figure 8.  BET and FGFR inhibitors suppress UM001 xenograft growth *in vivo*.**

A, B  (A) Nude mice injected subcutaneously with UM001 and LX-2 cells and (B) NSG mice injected with UM001 into the liver were treated with 90 mg/kg PLX51107, 5 mg/kg AZD4547, or the combination of PLX51107 and AZD4547. PLX51107 was given in chow form, and AZD4547 was given to mice by oral gavage. Data shown in (A) are mean ± SEM fold change in tumor volumes compared to average tumor volume at day 0. At day 23, statistical significance was measured between treatment groups, control ($n$ = 7), PLX51107 ($n$ = 5), AZD4547 ($n$ = 5) and PLX51107 + AZD4547 ($n$ = 6) using the unpaired *t*-test. (B) Representative CT scan images from each treatment group and average tumor sizes determined from the CT scans are shown. Tumor size is the maximum diameter of the tumor (green line in the CT scan images). The average tumor sizes at day 14 are mean ± SEM of data: control ($n$ = 3), PLX51107 ($n$ = 3), AZD4547 ($n$ = 3) and PLX51107 + AZD4547 ($n$ = 5). The unpaired *t*-test was used to calculate statistical significance.

compared to the FGFR4-specific inhibitor, BLU9931, indicating that FGFR1, 2, and/or 3 were the predominant receptors mediating FGF2 effects. In two separate *in vivo* models, we also identified that PLX51107 either increased or had little effect on UM001 tumor growth. This may indicate that the liver microenvironment including LX-2 cells plays a role in reducing the efficacy of BET inhibitors and co-inhibition of FGFRs by AZD4547 treatment significantly suppresses tumor growth compared to PLX51107-treated mice. These results suggest that co-targeting of BET and FGFRs is required to improve the responses of metastatic UM to BET inhibitors. Aside from FGFR inhibitors, histone deacetylase (HDAC) inhibitors have been shown to suppress FGF2-mediated upregulation of MMP gene expression and reverse FGF2-induced growth of human articular chondrocyte cultures (Wang *et al*, 2009). Although we found that FGF2 provides resistance to the HDAC inhibitor vorinostat, the combination of BET inhibitors and vorinostat induced a greater inhibition of UM001 growth compared to single BET inhibitor and vorinostat treatments (Appendix Fig S7). These findings suggest that testing of the combination of BET and HDAC inhibitors may be considered in *in vivo* models and clinically. Of note, vorinostat will be entering a Phase I clinical trial for metastatic UM patients (NCT03022565).

The majority of UM metastases are found in the liver and, hence, we focused on FGF2 secretion by HSCs which are capable of transdifferentiation into myofibroblasts and are implicated in the growth and progression of hepatic metastases (Kang *et al*, 2011). We determined that activated HSCs secrete FGF2, consistent with previous reports (Xu *et al*, 2005; Kang *et al*, 2011). The moderate resistance of OMM1.3 cells to BET inhibition following culture with LX2-conditioned media was also reversible by AZD4547, indicating involvement of FGFR1, 2, and/or 3 in mediating resistance. Although hepatocytes are the most abundant cell type in the liver (70–80%), we did not detect marked differences in FGF2 levels in media conditioned by hepatocytes compared to non-conditioned media (Appendix Fig S8A). This is not surprising as healthy hepatocytes are known to express minimal levels of FGFs (Sandhu *et al*, 2013). FGFs have also been shown to be produced by primary UM cell lines (Lefevre *et al*, 2009); however, we did not observe production of FGF2 by the metastatic UM cell lines used in this study (Appendix Fig S8B and C), which indicated that the autocrine FGF-FGFR activation loop may be cell type specific. Additionally, we found that BET inhibitor treatment of LX-2 cells increased their secretion of FGF2 as a potential adaptive response to BET inhibition. This finding is opposite to studies in other cancer types especially cancers with desmoplastic stroma such as pancreatic ductal adenocarcinoma (PDAC) where BET inhibitors were predominantly shown to decrease FGF2 expression and secretion of inflammatory cytokines from the TME as a mechanism inducing the anti-tumor effects of BET inhibitors (Yamamoto *et al*, 2016; Leal *et al*, 2017).

In addition to the increase in FGF2 production by LX-2, expression of FGFRs is also elevated in cell lines treated with BET inhibitors and importantly in patient tumor specimens following progression on PLX51107 treatment. Activation and/or increase in expression of RTKs has been shown in other cases of cancers (Chandarlapaty *et al*, 2011; Duncan *et al*, 2012; Kurimchak *et al*, 2016). Resistance to the JQ1 BET inhibitor in ovarian cancer cells has been associated with elevated expression of FGFRs (Kurimchak *et al*, 2016). The mechanism(s) underlying BET inhibitor-induced overexpression of FGFRs are unclear but may involve modulation of BRD4-induced regulation of *FGFR* transcription. BRD4 occupancy has been shown at the promoter region of RTKs, and this was attenuated by BET inhibitors (Stratikopoulos *et al*, 2015; Stuhlmiller *et al*, 2015).

In summary, findings from this study suggest that the elevated expression of FGFRs in patient tumors after PLX51107 treatment is a resistance mechanism in UM. While BET inhibitors are shown to be efficacious *in vitro* by inducing apoptosis and cell cycle arrest, we show evidence that FGF2 in the TME reduces the responses of tumor cells to BET inhibitor growth inhibitory effects. *In vivo* analysis of BET and FGFR inhibitor effects on UM xenografts show poor responses of tumors to BET inhibition, and this was reversed by a FGFR inhibitor. BET inhibitors also induce adaptive responses by upregulating FGF2 production by HSCs and FGFR expression in UM tumor cells. Hence, in patients with metastatic UM, it is likely that co-targeting of the FGF2/FGFR cascade is required to improve the efficacy of BET inhibitors.

# Materials and Methods

### Cell culture

UM001, UM004 and UM003 cells were derived from human UM metastases and were established at Thomas Jefferson University. OMM1.3 cells were obtained from Bruce Ksander's laboratory in 2014 (Yoshida *et al*, 2014; Cheng *et al*, 2015; Kageyama *et al*, 2017). UM001, UM004 and OMM1.3 were confirmed to harbor the Q209P mutation in *GNAQ* by Sanger sequencing; UM003 cells express the Q209L *GNAQ* mutation. The LX-2 HSC line was obtained from Dr. Scott L. Friedman in 2014 (Mount Sinai School of Medicine, New York, NY). All cell lines are confirmed mycoplasma-free. Culture conditions for UM001, UM004, UM003 and LX-2 cells are described in previous reports (Cheng *et al*, 2015, 2017). OMM1.3 cells were maintained in RPMI 1640 medium containing 10% FBS, 50 IU penicillin and 50 μg/ml streptomycin. For studies using the LX-2-conditioned media, media were collected from cultures after 3 days and then centrifuged for 4 min at 448 *g* to remove cell debris before ELISA or addition to UM cultures. Non-conditioned media control was media incubated without cells.

## Inhibitors, growth factors and antibodies

JQ1, AZD4547 and BLU9931 were purchased from Selleck Chemicals (Houston, TX). PLX51107 and PLX72853 were provided by Plexxikon, Inc. The structure of PLX51107 is reported previously (Ozer *et al*, 2018). All inhibitors were dissolved in DMSO. FGF2, IGF1, NRG1 and VEGF-A were purchased from Cell Signaling Technology (Danvers, MA). HGF, FGF1 and TGFα were from Peprotech (Rocky Hill, NJ). Cyclin A2 (C19) and SKP2 antibodies were from Santa Cruz Biotechnology, Inc. (Santa Cruz, CA). BimEL and Bmf (9G10) antibodies were purchased from Enzo Life Sciences (Farmingdale, NY). The cyclin D1 antibody was from BD Pharmingen (Franklin Lakes, NJ). The actin antibody was purchased from Sigma-Aldrich (St. Louis, MO). PLK1 (208G4), cyclin B1 (V152), p-Rb (S807/811), Wee1, p27, cleaved PARP (D214), c-Myc, Rad51 (D4B10), Hsp90 (C45G5), CDK1 (POH1), cyclin B1, Rb, Bad (D24A9), Puma, Bak, Bax, Bid, Bcl-2, Bcl-xL, FGFR1 (D8E4), FGFR2 (D4H9), FGFR3 (C51F2) and FGFR4 (D3B12) antibodies were purchased from Cell Signaling Technologies. Secondary antibodies were purchased from EMD Millipore (Burlington, MA).

## Growth assays

For crystal violet staining, cultures were rinsed with PBS and then incubated for 2 h with 0.2% (w/v) crystal violet in buffered (1:10 dilution) formalin. High magnification images of crystal violet staining were obtained using the Nikon Eclipse Ti-E microscope (Nikon, Japan), and percentage area covered by stain was quantitated using ImageJ.

## Annexin V/propidium iodide (PI) staining

Cells were trypsinized, and cell pellets were washed twice with PBS and then re-suspended in binding buffer containing 1:20 annexin V-APC (BD Pharmingen) and 0.2 mg/ml PI (Life Technologies; Carlsbad, CA). Samples were incubated at room temperature for 15 min, protected from light, and annexin V-APC and PI fluorescence analyzed by flow cytometry.

## EdU incorporation assay

Cells cultured at $4 \times 10^5$ cells/well in 6-well plates were treated with BET inhibitors and/or FGF2 for 48 h. Sixteen hours prior to the end of treatment, 10 μM EdU was added to cultures and then cells processed using the Click-iT™ Plus EdU Alexa Fluor™ 594 flow cytometry assay kit (Life Technologies) according to manufacturer's protocol.

## Western blotting

Cells were washed twice with PBS and lysed into Laemmli sample buffer (Bio-Rad, Hercules, CA) containing β-mercaptoethanol. Cell lysates were heated at 95°C for 10 min. Proteins were separated by SDS–PAGE and transferred onto PVDF membranes. Membranes were blocked for 90 min and then incubated with primary antibody [in 1% BSA/PBS containing Tween®-20 (PBST)] overnight at 4°C. All primary antibodies except SKP2 and actin antibodies were diluted 1:1,000. The SKP2 antibody was diluted 1:250, and the actin antibody was diluted 1:3,000. Membranes were washed 3 × 10 min with PBST and incubated for 90 min at room temperature with secondary antibody (1:2,500 in 1% skimmed milk/PBST). After a final wash, protein expression was determined by chemiluminescence on the ChemiDoc (Bio-Rad).

## RPPA

Cells were lysed and prepared as described previously (Tibes *et al*, 2006) and RPPA performed at the MD Anderson Functional Proteomic core facility (Houston, TX). RPPA data were determined and analyzed as described (Cheng *et al*, 2017). Comparisons were performed between conditioned samples using the two-sample *t*-test method with 1,000 permutations. Multiple hypothesis test corrections were calculated, and antibodies with a Storey *q*-value < 0.05 and a fold ratio > 2 were considered significant, unless noted otherwise. Calculations were performed in MATLAB® (v2017b) using the mattest and mafdr functions.

## Enzyme-linked immunosorbent assay (ELISA)

Media conditioned by LX-2, UM001 and UM004 cells were collected for analysis of FGF2 levels using the human basic FGF ELISA kit (Abcam, UK). Human hepatocyte (HH)-conditioned media was assessed using the Human FGF basic Quantikine ELISA kit (R&D Systems; Minneapolis, MN). Samples were processed according to manufacturer's protocol. FGF2 concentration was extrapolated from the standard curve generated using standards in the kit.

## Patient tumor samples

Biopsies were collected from metastasis of UM patients enrolled in the PLX51107 clinical trial (NCT02683395). Samples were collected under an IRB-approved protocol (IRB#: 02.9014R) that included written informed consent and was in accordance with recognized ethical guidelines. Experiments performed with the biopsies conform to principles set out in the WMA Declaration of Helsinki and the Department of Health and Human Services Belmont Report. The tissue samples were fixed in buffered formalin (1:10) for 24 h, then embedded in paraffin, and sections were stained for FGFRs by immunohistochemistry (IHC). Antibodies against FGFR1 (D8E4) and FGFR2 (D4H9) were from Cell Signaling Technologies, whereas the FGFR4 antibody (ab5481) was purchased from Abcam.

## *In vivo* tumor studies

Animal experiments were performed at the Thomas Jefferson University animal facility that is accredited by the Association for the Assessment and Accreditation of Laboratory Animal Care and has a full-time veterinarian. Mice cages were limited to 2–4 mice per cage and checked daily for cage cleanliness and sufficient water. Food/chow was checked or re-filled at least three times a week. *In vivo* studies were approved by the Institutional Animal Care and Use Committee (IACUC). Male athymic nude (*nu/nu*; homozygous, 6–8 weeks) and NSG (female, 7 weeks) mice were purchased from The Jackson Laboratory (Bar Harbor, ME). For subcutaneous

## The paper explained

### Problem

Bromodomain and extraterminal inhibitors are shown to be effective pre-clinically in suppressing tumor growth including UM. However, there is evidence showing progression of hepatic metastases in UM patients following PLX51107 BET inhibitor treatment and mechanisms of resistance to BET inhibitors in UM are not known.

### Results

We showed that FGF2 secreted from stromal cells in the liver microenvironment provided protection against the growth inhibitory effects of BET inhibitors in metastatic UM cells. FGF2-induced resistance to BET inhibition was reversible by FGFR inhibitors. In addition, we identified that BET inhibitors increased FGF2 secretion by hepatic stellate cells. In both UM cell lines and patient tumors, BET inhibitor treatment elevated the expression of FGFRs. We also showed that responses of xenograft tumors to a BET inhibitor were poor *in vivo* but this is reversed by FGFR inhibitors.

### Impact

Our findings identified a mechanism that is mediating the poor responses of UM to BET inhibitors and provided a strategy to improve the efficacy of BET inhibitors in these patients.

injection of nude mice, a 100 μl mixture of $5 \times 10^6$ UM001 and $5 \times 10^6$ LX-2 cells was injected into mice. Once tumor xenografts were established (100–200 mm$^3$), the animals were divided into four arms: control ($n = 8$), PLX51107 (90 mg/kg, $n = 8$), AZD4547 (5 mg/kg, $n = 8$), or the combination of PLX51107 and AZD4547 ($n = 10$). Tumor volume was calculated by digital caliper measurements and the formula: volume = length × (width$^2$/2). For the liver orthotopic model, a 20 μl mixture of $1 \times 10^6$ UM001 cell suspension and Matrigel (BD Biosciences, Bedford, MA; 2:1 ratio) was injected into the liver of NSG mice. After 6–8 weeks, mice were divided into: control ($n = 6$), PLX51107 ($n = 6$), AZD4547 ($n = 6$), or the combination of PLX51107 and AZD4547 ($n = 6$) arms. Tumor size (maximum tumor diameter) in the liver was determined by computerized tomography (CT). PLX51107 chow was given to mice continuously while AZD4547 was fed to mice by oral gavage once daily in the following pattern: 5 days on-drug and 2 days off-drug. AZD4547 was formulated in 100% DMSO. Mice not treated with AZD4547 were fed by oral gavage with 100% DMSO. Mouse weight was monitored thrice a week (Fig EV5).

### Statistical analysis

For all quantitative *in vitro* data, experiments were repeated 3–6 times and the mean and standard error of the mean (SEM) of data are calculated from at least triplicate experiments. Outliers that were more than twofold different compared to other datasets were excluded from analysis. For *in vivo* tumor data, the experiments commenced with at least six animals per treatment group. Data from animals with no tumors formed prior to the experiment or had died during the experiment were excluded from analysis. The mean and SEM of data at the final treatment timepoint were calculated from at least three animals. The Shapiro–Wilk test was used to check normality of sample distribution for all experiments, and based on results from this test, statistical significance was calculated using the unpaired *t*-test.

Expanded View for this article is available online.

## Acknowledgements

We thank Dr. John L. Farber and Dr. Peter A. McCue of Thomas Jefferson University Hospital for pathological expertise on patient samples, and Plexxikon, Inc., for providing PLX51107 and PLX72853. The authors thank Dr. Michael Davies, Dr. Wanleng Deng, and the MD Anderson Cancer Center Functional Proteomic core facility for performing RPPA. The authors also thank the Sidney Kimmel Cancer Center flow cytometry facility and pathology core for assistance with flow cytometry and immunohistochemistry, respectively. This project is funded by a Melanoma Research Alliance (MRA) team science award (PI Gary Schwartz). Dr. Aplin is also supported by a Cure Ocular Melanoma/Melanoma Research Foundation Established Investigator award and the Dr. Ralph and Marian Falk Medical Research Trust Bank of America, N.A., Trustee. Dr. Chua and her research are also supported by the AACR-Ocular Melanoma Foundation Fellowship and a National Cancer Center post-doctoral fellowship. The PLX51107 clinical trial is funded by Plexxikon, Inc. RPPA studies are supported by the Dr. Miriam and Sheldon G. Adelson Medical Research Foundation.

## Author contributions

VC designed the study, performed most experiments including analysis, and wrote the manuscript. CL assisted with cell growth, Western blotting, and flow cytometry assays. JLFT helped with the *in vivo* experiment involving the subcutaneous model. TSu performed the liver orthotopic experiment with assistance from MT and BQL. This experiment was supervised by TSa. TJP analyzed RPPA data and generated heatmaps. MO provided patient tumor samples from the PLX51107 clinical trial and patient data. AEA provided guidance and feedback on the experimental design of this study and writing of this manuscript. GA, GS, RDC, and TSa provided expert feedback. All authors critically read this manuscript.

## Conflict of interest

A.E. Aplin reports receiving a commercial research grant from Pfizer and has ownership interest in patent number 9880150.

## For more information

(i) Ocular Melanoma Foundation (http://www.ocularmelanoma.org/)

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
