## [Review Process File · EMBO Molecular Medicine]

Stromal Fibroblast Growth Factor 2 Reduces the Efficacy of Bromodomain Inhibitors in Uveal Melanoma

Vivian Chua, Marlana Orloff, Jessica LF Teh, Takahito Sugase, Connie Liao, Timothy J. Purwin, Bao Q Lam, Mizue Terai, Grazia Ambrosini, Richard D. Carvajal, Gary Schwartz, Takami Sato, Andrew E. Aplin

Review timeline:

Submission date:	6 March 2018
Editorial Decision:	9 March 2018
Correspondence:	13 March 2018
Resubmission:	12 June 2018
Editorial Decision:	6 July 2018
Revision received:	29 October 2018
Editorial Decision:	21 November 2018
Revision received:	30 November 2018
Accepted:	11 December 2018

Editor: Lise Roth

Transaction Report:

1st Editorial Decision

9 March 2018

Thank you for submitting your manuscript to EMBO Molecular Medicine. I have now had a chance to read your research article carefully and to discuss it with the other members of our editorial team. I am sorry to inform you that we find that the manuscript is not well suited for publication in EMBO Molecular Medicine and that we therefore have decided not to proceed with peer review.

Your study investigates resistance to BET inhibitor therapy in the context of uveal melanoma (UM). A next-generation BET inhibitor, PLX51107 inhibited UM cell growth and induced apoptosis, which was rescued by FGF2. FGF2 was detected in conditioned medium collected from activated stellate cells, and its secretion was increased upon BET inhibitor treatment. In biopsies from a UM patient, FGFR2 expression was enhanced post-treatment, confirming an adaptive response. Targeting of FGFRs in UM cancer cells improved BET inhibitor responses in vitro.

We appreciate that your data suggest that co-targeting of FGFR signaling and BET proteins might maximize the response of metastatic uveal melanoma to BET inhibitors. However, the in vivo translational applications of your work are not further developed here, thereby limiting the overall translational and clinical insights that are key for publication in EMBO Molecular Medicine. Therefore, I am afraid that we cannot offer further consideration to your article.

Please rest assured that this is not a judgment of the quality or interest of your work but a decision based on appropriateness for EMBO Molecular Medicine.

I am sorry that we cannot be more positive on this occasion.

Correspondence

13 March – 11 June 2018

AUTHOR: Thank you for your comments. Obviously, we are disappointed in the outcome. The acquisition of uveal melanoma samples from liver metastases pre- and post treatment for analysis of resistance mechanisms is unique. However, we appreciate that reviewers may expect in vivo studies since our studies would inform on-going clinical trials with PLX51107. I am wondering about options if we performed an in vivo experiment.

EDITOR: We do appreciate your work on uveal melanoma samples, but as mentioned in the decision letter, we would need in vivo data to assess the translational applications of your findings. Should you include additional in vivo data to strengthen the translational value of your work, we would be happy to consider it for publication.

AUTHOR: We have performed in vivo experiments to test the combination of BETi and FGFR inhibition and intend to resubmit the manuscript. We hope that you will re-consider the manuscript for EMBO Molecular Medicine.

2nd Editorial Decision

6 July 2018

Thank you for the submission of your manuscript to EMBO Molecular Medicine. We have now heard back from the three referees whom we asked to evaluate your manuscript.

As you will see from the reports below, while the referees all mention the interest, novelty and clinical importance of the study, they also agree that strengthening of the data to fully support the conclusions will be necessary for publication in EMBO Molecular Medicine. Our cross-commenting exercise helped clarifying the most critical points to address during revision, and we would encourage you to focus on the following:

- An appropriate tumor model is essential to support the conclusions and address the role of the hepatic niche (injection in the liver capsule and/or in the spleen).
- If possible, more patients' samples should be added, however referees are aware that this might be difficult to achieve. This should be discussed.
- Experiments with HDAC inhibitors are suggested, but not essential for the main message of the manuscript
- Other points to address include addition of appropriate controls, justification on the different drugs' mode of administration and compounds characterization, extended statistics, homogenization of the cell lines used in all experiments, characterization of inflammatory markers, and use of additional markers of programmed cell death.

Addressing the above reviewers' concerns in full, and experimentally as needed, will be necessary for further considering the manuscript in our journal. EMBO Molecular Medicine encourages a single round of revision only and therefore, acceptance or rejection of the manuscript will depend on the completeness of your responses included in the next, final version of the manuscript.

Please also contact us as soon as possible if similar work is published elsewhere. If other work is published, we may not be able to extend the revision period beyond three months.

I look forward to receiving your revised manuscript.

***** Reviewer's comments *****

Referee #1 (Comments on Novelty/Model System for Author):

The authors use a xenograft model by subcutaneous inoculation of cells. Wish they tried an orthotopic model so they could recapitulate natural course of disease, as this could affect the microenvironment-tumor interactions and an array of chemokines.

Referee #1 (Remarks for Author):

Despite being the most common intraocular malignancy in adults, there is no therapies for the metastatic form of uveal melanoma. This in part, due to the development of resistance to targeted therapies. The manuscript from Chua et al is a well written manuscript highlighting an unmet need of the field.

A disconnection exists between pre-clinical assessment of drugs and their clinical outcomes. Chua and colleagues identified a mechanism mediating poor responses of UM to BET inhibitors. The following questions need to be answered by the authors:

1. What are the half-lives of the compounds used in the experiments? For instance, JQ1 has been reported to have a short half life in plasma (0.9hrs; Filippakoupoulos et al, Nature, 2010).
2. Were any inflammatory markers assessed?
3. Why UM003 is not used in all experiments, as the other cell lines? This is important as this cell line has a different mutation.
4. Although proteins associated with cell cycle were evaluated by Western blot analyses, why there is no classic cell cycle studies involving PI labeling? Why using EdU incorporation as an S-phase marker? This could have been done with PI labeling and be clearer.
5. What were the concentrations used for the growth factors? Are these physiologically relevant?
6. For the in vivo experiments, did you evaluate inflammatory markers? Can you please show a representative image of the tumor size?
7. Please include in the Materials and Methods the concentrations of the antibodies used.
8. What was the final volume of the inoculated mixture of UM001 and LX-2 cells?
9. The PLX51107 was administered through the chow, while AZD4547 was fed orally. Was the absorption of drugs similar? What studies were previously done to ensure this?

Referee #2 (Remarks for Author):

In the present manuscript, which this reviewer sees for the first time, the authors show that fibroblast growth factor 2 (FGF2) rescues UM cells from growth inhibition by BET inhibitors in vitro. FGF2 effects are reversible by FGF receptor (FGFR) inhibitors. BET inhibitors also increase FGFR protein expression in UM cells and in patient samples. Hepatic stellate cells (HSCs) secrete FGF2 with HSC conditioned media shown to provide resistance of UM cells to BET inhibitors. Interestingly, PLX51107 increased in vivo tumor growth of UM cells co-injected into mice with HSCs and the combination of PLX51107 and FGFR inhibitor, AZD4547, suppressed tumor growth. Thus the authors suggest that co-targeting of FGFR signaling is required to increase the responses of metastatic UM to BET inhibitors.

The authors indicate that BET inhibitors significantly increased apoptosis, but data are mainly relative to Annexin V and PARP cleavage. Many more markers of the different kinds of apoptosis (or other programmed cell deaths) might be included to strengthen the analyses.

The authors indicate that FGF2 rescues metastatic UM cells from the growth inhibitory effects of BET inhibitors. It has been shown that other chromatin acting drugs, such as HDACi, strongly antagonize FGF2 effects (doi: 10.1080/08977190802625179; doi: 10.1167/iavs.09-4538). The authors may want to include HDACi alone or in combination with BETi and or FGF2 inhibitors to provide a more (possibly more effective) combo treatment. It is important to underline that HDACi might likely quickly be included in treatment since they are in clinical trials and approved for different cancer conditions. Along these lines, HDAC inhibitors induced morphologic differentiation, cell cycle exit, and a shift to a differentiated, melanocytic gene expression profile in cultured UM cells. HDACi inhibited the growth of UM tumors in vivo, suggesting that a combo treatment should be beneficial also in vivo (doi: 10.1158/1078-0432.CCR-11-0946). The

combination of a histone deacetylase inhibitor (Entinostat) and pembrolizumab is also currently under investigation in a phase II trial (NCT02697630) in Metastatic Uveal Melanoma (DOI: <https://doi.org/10.17925/OHR.2017.13.02.100>), suggesting that experiments into this direction (combo treat) should be performed both in vitro and in vivo.

The authors suggest that elevation of FGFR signals represents as an adaptive response of UM to BET inhibitor. This hypothesis is supported by data on cell lines and one patient sample before and after BETi treatment. It might be of interest, if possible, to include additional patient samples.

The in vivo data have been obtained using UM001 and LX-2 cells. It would be much more important to include if possible, patient 'derived xenotransplants. In addition, molecular data from mice tumors in different stages of the treatment should be presented in the study.

Referee #3 (Comments on Novelty/Model System for Author):

The authors applied both in vitro and in vivo systems. To study drug effects of BET inhibitors on uveal melanoma cells colony formation assays were performed. To delineate the role of the hepatic microenvironment on tumor cells and resistance development the media was altered by adding growth factors including FGF2 or hepatic stellate cell (HSC)-conditioned media. This nicely provides evidence of interaction of HSC with melanoma cells driving resistance to BET inhibitors.

However, approaches are missing to delineate this cellular cross-talk. In the in vivo model both LX-2 and UMM01 cells were co-injected subcutaneously. This does not recapitulate the organ-specific hepatic microenvironment. Injection of the uveal melanoma cells into the liver capsule to simulate hepatic metastatic disease should be performed which has already been shown by the authors in a previous study (Cheng H, *Canc Res*, 2015). Besides, the liver could be colonized with uveal melanoma cells by spleen injection. Studying drug effects in such in vivo models would be far more relevant than subcutaneous injections. Moreover, in such experimental settings the biological relevance of the FGF2/FGFR cross-talk on resistance to BET inhibitors could be studied thoroughly.

Referee #3 (Remarks for Author):

Chua and colleagues studied resistance mechanisms developing during BET inhibitor therapy against uveal melanoma due to a highly interesting clinical case of hepatic uveal melanoma metastasis that progressed upon BET inhibition. In vitro, the use of BET inhibitors was found to increase apoptosis of several human uveal melanoma cells. This could be rescued by addition of FGF2 that was shown to be released by activated hepatic stellate cells (HSCs). In a murine xenograft model a reduction of uveal melanoma growth was observed when the resistance to a BET inhibitor was diminished by administration of an additional pan-FGFR1/2/3 inhibitor. Altogether, the authors describe HSCs to be the responsible cell compartment in the hepatic niche that mediates resistance to BET inhibition by secretion of FGF2.

The overall concept of the study is well-designed and of great interest. It can be seen as example of translational medicine. However, there are certain limitations that I would like to emphasize.

- One major point is the in vivo model that was used to validate the results of the in vitro studies. In athymic mice (nude mice) one human uveal melanoma cell line was co-injected with an activated HSC cell line subcutaneously, which should represent the interaction of HSCs and uveal melanoma cells in the hepatic niche. However, the hepatic niche is a highly complex niche consisting of other cells like for example liver sinusoidal endothelial cells, Kupffer cells and hepatocytes. Therefore, this study would benefit from injection of uveal melanoma cells into the liver capsule (as performed in a previous study by the authors - Cheng H, *Canc Res*, 2015). Moreover, cells can be injected into the spleen to colonize the liver. After establishment of the uveal melanoma cells in the hepatic niche drug combinations and its effects on metastases should be validated. Such an approach would represent the clinical case with hepatic metastasis far better than a subcutaneous injection and is needed as proof of principle.

- Secondly, the statistical analysis should be extended. The authors just "assumed unequal variance". This should be verified before applying the corresponding statistical tests. Please check for normality of your sample distribution.

Minor points/questions I would like to address are expressed below.

- Data from one patient with progress of hepatic metastasis is presented. Do you have data on progression of metastatic disease in other cases, as well? This would really help to tie the effect to the administration of PLX51107.
- Why did you choose the panel of FGF2, HGF, IGF1 and NRG1 as hepatic growth factors? Do you have other data indicating that these are the ones responsible for HSC interaction with uveal melanoma cells? There are plenty of factors in the hepatic microenvironment, released by hepatocytes or even liver sinusoidal endothelial cells.
- FGF2 did not rescue OMM1.3 cells. A trend is seen here. However, this was not statistically significant. This should be repeated please.
- Why do you think effects in OMM1.3 cells are generally weaker than in the UM001, UM003 and UM004?
- Figure 5/6: Controls are missing here. It would be interesting to see the effect of single administration of either AZD4547 or BLU9931 on the uveal melanoma cell lines as well as in combination with the BET inhibitors.
- Uveal melanoma cells have only been co-injected with LX-2 cells. Why did you not inject them alone? This would be an important control.
- Why did you only use UM001 cells for the in vivo experiments? In the in vitro studies other uveal melanoma cell lines were applied, as well. The quality of the study would benefit from repetition and verification of this important experiment with other cell lines.
- Besides, in the in vivo system there is a discrepancy during drug administration. BET inhibitors were administered every day by food uptake. However, the FGFR receptor, AZD4547, was administered by oral gavage on a "5 days on, 2 days off" scheme. Why did you not give AZD4547 every day like you did with PLX51107?
- Page 12: No difference in FGF2 levels in media conditioned by hepatocytes was described. However, data is not shown. Please provide evidence.

Overall, the manuscript is written very clearly. There is just a minor comment. On page 16 there is a typo ("." before "whereas").

Prognosis of uveal melanoma patients with metastatic disease is very poor. This study provides an example how to delineate the interaction of organ-specific cells of the liver with tumor cells. For future therapies research approaches like this are of great importance to fight hepatic metastatic disease.

1st Revision - authors' response

29 October 2018

Response to Reviewers' Comments

We thank the reviewers for their comments which we have addressed below. We believe that these revisions have strengthened this manuscript.

Referee #1

1) Wish they tried an orthotopic model so they could recapitulate natural course of disease, as this could affect the microenvironment-tumor interactions and an array of chemokines.

Response: We agree that an orthotopic model will better recapitulate the disease in the presence of appropriate microenvironment-tumor interactions. Hence, we have performed the liver injection model which involves injecting UM001 directly to the liver of NSG mice and determined effects of BET and FGFR inhibitors on xenograft tumor growth in the liver. Results are shown in Fig 8B.

2) What are the half-lives of the compounds used in the experiments? For instance, JQ1 has been reported to have a short half life in plasma (0.9hrs; Filippakopoulos et al, Nature, 2010).

Response: Based on previous reports, PLX51107 has a half-life of ~2.8 hours in plasma (Ozer et al., 2018) whereas JQ1 has a half-life of 0.9-1.4 hours (Filippakopoulos et al., 2010). This information is added to the manuscript.

3) Were any inflammatory markers assessed?

Response: We have now assessed expression of inflammatory markers. As BET inhibitors such as JQ1 have been shown to induce anti-inflammatory responses and suppress transcription of inflammatory cytokines (Jahagirdar et al., 2017; Meng et al., 2014), we investigated PD-L1, IL6 and

IL1B expression following BET inhibitor treatment of UM001. We found that PLX51107 markedly reduces the expression of IFN γ -induced PD-L1 and moderately decreases *IL6* mRNA levels (Figure A). *IL1B* expression was undetected in UM001 (data not shown).

Figure A: Effects of BET inhibition and FGF2 on expression levels of (i) PD-L1 and (ii) IL6 in UM001.

4) Why UM003 is not used in all experiments, as the other cell lines? This is important as this cell line has a different mutation.

Response: We have now performed crystal violet growth assays investigating effects of the FGFR1/2/3 inhibitor, AZD4547, in reversing FGF2-induced resistance to BET inhibitors in UM003 cells (Fig EV6). Similar to the other uveal melanoma cell lines (Fig 5), AZD4547 significantly reduces FGF2-induced resistance to BET inhibition in UM003 (Fig EV6).

In addition, we used the probabilistic machine learning method PolyPhen-2 to predict the effects of all possible SNP mutations that would cause a missense mutation at the GNAQ p.Q209 sequence location. The predicted mutational effect results produced from PolyPhen-2 show that both GNAQ Q209L and Q209P are ‘deleterious’ and have almost identical probabilistic damaging alteration scores (Figure Bi). Based on these results we do not believe there will be any difference between GNAQ p.Q209L and p.Q209P mutants. We have also performed differential expression analysis of genes in the TCGA samples comparing GNAQ Q209P or Q209L to GNAQ WT (Figure Bii). We observe a similarity in gene expression profiles between GNAQ Q209P and Q209L samples.

Figure B: (i) Prediction of effects of GNAQ Q209P and Q209L using PolyPhen-2. (ii) Heatmap of gene expression in GNAQ Q209P, Q209L and WT data sets in UM TCGA.

5) Although proteins associated with cell cycle were evaluated by Western blot analyses, why there is no classic cell cycle studies involving PI labeling? Why using EdU incorporation as an S-phase marker? This could have been done with PI labeling and be clearer.

Response: EdU incorporation assays were performed to determine DNA synthesis or progression to S-phase. As shown in Fig 4A, BET inhibitors decreased EdU incorporation and this is reversed

significantly by FGF2, indicating that BET inhibitor mediated decrease in DNA synthesis was rescued by FGF2. However, we have obtained PI labeling results to determine changes in cells progressing through the cell cycle. Results are shown for UM001 cells in Fig EV4. We observed that BET inhibitors increased % cells in sub-G1, an effect which was significantly reversed by FGF2. We did not see marked changes in % cells in G0/G1 and S phase but we observed a moderate rescue by FGF2 of BET inhibitor-induced decrease in % cells in G2/M phase (Fig EV4). We have included these additional results to the manuscript.

6) What were the concentrations used for the growth factors? Are these physiologically relevant?

Response: The concentration used for growth factors was 50ng/mL. Plasma or serum levels of FGF2 have been reported to be in the pg/mL range (Larsson et al., 2002) but in cancers and following invasion of tumor cells to the liver, FGF2 levels can increase (Cronauer et al., 1997; Jinno et al., 1997; Sato et al., 2002). Therefore, in addition to testing exogenous FGF2 effects, we collected stellate cell conditioned media to determine whether endogenous FGF2 rescues BET inhibitor effects. As shown in Fig 6A, we observed approximately 8ng/mL FGF2 in LX-2 stellate cell conditioned media which provided rescue of BET inhibition in OMM1.3. In addition, we have also tested increasing concentrations of FGF2 from 1ng/mL to 50ng/mL and identified a gradual increase in UM001 growth in the presence of BET inhibitors (Fig EV2).

7) For the *in vivo* experiments, did you evaluate inflammatory markers? Can you please show a representative image of the tumor size?

Response: As we have addressed in comment 3, we have now investigated PD-L1, IL6 and IL1B expression levels in UM001 and observed downregulation of PD-L1 and IL6 following PLX51107 treatment. We investigated UM001 xenograft tumors we collected from the *in vivo* assay (subcutaneous injection model). We were not able to detect PD-L1 in the absence of IFN γ induction (data not shown) but we observed moderate downregulation of *IL6* mRNA levels (Figure Ci) which was similar to results shown *in vitro* (Figure A). A representative image of the tumor size is shown in Figure Cii.

Figure C: (i) *IL6* mRNA levels following PLX51107 treatment of xenograft UM001 tumors. (ii) Representative image of UM001 xenograft tumor.

8) Please include in the Materials and Methods the concentrations of the antibodies used.

Response: Details on antibody concentrations have now been added to Materials and Methods.

9) What was the final volume of the inoculated mixture of UM001 and LX-2 cells?

Response: The final volume of the mixture is 100 μ L. We have added this information to the materials and methods section.

10) The PLX51107 was administered through the chow, while AZD4547 was fed orally. Was the absorption of drugs similar? What studies were previously done to ensure this?

Response: Pharmacokinetics of PLX51107 and AZD4547 have been reported previously (Gavine et al., 2012; Ozer et al., 2018). When fed orally, PLX51107 and AZD4547 have similar absorption rates with the plasma concentrations of both drugs shown to peak at 1h. However, the half life of PLX51107 ($t_{1/2}$: 2.8h) is shorter than AZD4547 ($t_{1/2}$: 10h). We also see that the plasma concentration of PLX51107, when fed as chow, in mice was consistently at ~1000ng/mL at multiple time points during the day (data not shown).

Referee #2

1) The authors indicate that BET inhibitors significantly increased apoptosis, but data are mainly relative to Annexin V and PARP cleavage. Many more markers of the different kinds of apoptosis (or other programmed cell deaths) might be included to strengthen the analyses.

Response: We have explored other apoptotic and anti-apoptotic markers (Fig EV5). We identified that in all three cell lines, UM001, UM004 and OMM1.3, BET inhibitors markedly increased expression of pro-apoptotic proteins, Bim-EL and Bmf, and downregulated expression of Bid which has been reported to be an anti-apoptotic protein (Luo et al., 2010). The results on Bid have been included in the manuscript results section.

2) The authors indicate that FGF2 rescues metastatic UM cells from the growth inhibitory effects of BET inhibitors. It has been shown that other chromatin acting drugs, such as HDACi, strongly antagonize FGF2 effects (doi: 10.1080/08977190802625179; doi: 10.1167/iavs.09-4538). The authors may want to include HDACi alone or in combination with BETi and or FGF2 inhibitors to provide a more (possibly more effective) combo treatment.

Response: In the publication by Wang et al., trichostatin A (TSA), a HDAC inhibitor, was shown to inhibit FGF2-induced upregulation of *MMP1* and *MMP13* gene expression but not *ADAMTS5* levels (Wang et al., 2009). Interestingly, the authors also showed that TSA did not significantly alter cell numbers but increased cell death. TSA reversed FGF2-induced increase in cell numbers and FGF2 blocked cell death in TSA-treated cultures. This year, vorinostat (HDAC inhibitor) will be tested in a Phase I clinical trial for metastatic UM patients. Hence, we have investigated effects of vorinostat on FGF2 and BET inhibitor treated UM cultures to determine whether HDAC inhibitors reverse FGF2-mediated resistance to BET inhibitors. In the absence and presence of FGF2, the combination of vorinostat and BET inhibitors led to a greater inhibition of growth of UM001 compared to single vorinostat and BET inhibitor treatment (Figure D). We think that co-targeting of BET and HDAC will be considered for in vivo and clinical studies but at present and in this manuscript, as FGF2 binds canonically to FGFRs and BET inhibitors upregulate FGFR expression in cell lines and patient tumors, we focused on testing FGFR inhibitors. However, we have discussed about HDAC

inhibitors in the manuscript.

Figure D: UM001 was treated with FGF2 and BET inhibitors +/- vorinostat. Changes in colony growth were detected by crystal violet staining.

3) The authors suggest that elevation of FGFR signals represents as an adaptive response of UM to BET inhibitor. This hypothesis is supported by data on cell lines and one patient sample before and after BETi treatment. It might be of interest, if possible, to include additional patient samples.

Response: Since submitting this manuscript for review, we have received pre- and post-treatment tumor biopsies from another patient (patient #3) in the BET inhibitor trial. We stained the samples for FGFRs. We found that in this patient, we observed increased FGFR1 staining in tumor cells in the post-treatment biopsy compared to the pre-treatment biopsy. This result is added to Fig 7C and details on patient's treatment history and MRI scans are shown in Fig 1.

During the revision, the data for patient #2 underwent a second independent pathologist review that raised concerns about the tumor selectivity of FGFR staining. We have concluded that staining for FGFR1 and 4 were negative in the tumor cells and no marked changes were seen in FGFR2 staining between pre- and post-treatment samples. Hence, these data are transferred to Fig EV9 and clarified in the manuscript. Overall, we believe that a strength of our study is the use of biopsies from liver metastases. We hoped to receive biopsy specimens from more patients to support our data but these samples are hard to obtain, hence, we are only able to include data from two patients in this manuscript.

4) The in vivo data have been obtained using UM001 and LX-2 cells. It would be much more important to include if possible, patient 'derived xenotransplants. In addition, molecular data from mice tumors in different stages of the treatment should be presented in the study.

Response: PDX models have been shown by other research groups to grow slowly in vivo and there are limited uveal melanoma patient samples. Therefore, we have not included PDX models. However, we have performed the liver orthotopic model with UM001 (Fig 8B). For molecular data, we have collected UM001 tumor xenografts (subcutaneous injection model) at multiple time points of BET and FGFR inhibitor treatment. We investigated BIM and cleaved PARP. Bim levels were upregulated more rapidly in tumors of PLX51107 and AZD4547 treated mice compared to PLX51107-treated and control mice (Figure E). Cleaved PARP protein levels were not detected by western blotting in control xenograft tumors, PLX51107 increased cleaved PARP level at day 14 and the combination of PLX51107 and AZD4547 increased cleaved PARP expression from day 2 of treatment (Figure E). These results show that the combination of both agents induced a more rapid upregulation of apoptosis in the xenograft tumors.

Figure E: Bim and cleaved PARP protein expression over 14 days of treatment of mice bearing UM001 xenografts with PLX51107 and AZD4547.

Referee #3

1) In the in vivo model both LX-2 and UMM01 cells were co-injected subcutaneously. This does not recapitulate the organ-specific hepatic microenvironment. Injection of the uveal melanoma cells into the liver capsule to simulate hepatic metastatic disease should be performed.

2) After establishment of the uveal melanoma cells in the hepatic niche drug combinations and its effects on metastases should be validated. Such an approach would represent the clinical case with hepatic metastasis far better than a subcutaneous injection and is needed as proof of principle.

Response: In collaboration with Dr. Takami Sato, we have performed the liver injection model. Results are shown in Fig 8B.

3) The statistical analysis should be extended. The authors just "assumed unequal variance". This should be verified before applying the corresponding statistical tests. Please check for normality of your sample distribution.

Response: We have performed the Shapiro-Wilks test to determine normality of samples. Our data passed the test and hence, we performed the unpaired t-test for statistical significance.

4) Data from one patient with progress of hepatic metastasis is presented. Do you have data on progression of metastatic disease in other cases, as well? This would really help to tie the effect to the administration of PLX51107.

Response: Please see response to comment #3 by referee 2. FGFR staining results from patient #3 is shown in Fig 7C and data from patient #2 is transferred to Fig EV9.

5) Why did you choose the panel of FGF2, HGF, IGF1 and NRG1 as hepatic growth factors? Do you have other data indicating that these are the ones responsible for HSC interaction with uveal melanoma cells? There are plenty of factors in the hepatic microenvironment, released by hepatocytes or even liver sinusoidal endothelial cells.

Response: The growth factors that were investigated in Fig 3A are widely known to be liver-secreted factors (Bohm et al., 2010). However, we have explored a number of other growth factors including VEGF-A, FGF1 and TGF- α , and showed that VEGF-A and TGF- α do not rescue BET inhibitor effects in UM. FGF1 moderately rescued BET inhibitor-mediated growth inhibition in UM001 and effects were weaker than FGF2. These results have been added to Fig EV3.

6) FGF2 did not rescue OMM1.3 cells. A trend is seen here. However, this was not statistically significant. This should be repeated please.

Response: We repeated the experiments with OMM1.3 multiple times but we consistently observe that FGF2 do not significantly rescue BET inhibitor effects in OMM1.3 cultures.

7) Why do you think effects in OMM1.3 cells are generally weaker than in the UM001, UM003 and UM004?

Response: At this stage, we are unclear as to why effects in OMM1.3 cells are generally weaker. We have repeated experiments with OMM1.3 multiple times and consistently, results were not significant. An extensive genetic and/or proteomic profiling of the cell lines may uncover causes for differences between the cell lines. In addition, we think that the low basal FGFR expression and lack of increase in FGFR protein levels following BET inhibitor treatment in OMM1.3 compared to other cell lines may lead to weaker responses of OMM1.3 to FGF2 (Fig 7B).

8) Figure 5/6: Controls are missing here. It would be interesting to see the effect of single administration of either AZD4547 or BLU9931 on the uveal melanoma cell lines as well as in combination with the BET inhibitors.

Response: We have now added controls which are shown in Fig EV8. Single administration of AZD4547 or BLU9931 at 1 μ M has little effects on UM001, UM004 and OMM1.3 cell viability but when added in combination with BET inhibitors and FGF2, the FGFR inhibitors at this concentration reduce FGF2-mediated rescue of BET inhibitors.

9) Uveal melanoma cells have only been co-injected with LX-2 cells. Why did you not inject them alone? This would be an important control.

Response: LX-2 cells do not form into tumors in vivo ((Amann et al., 2009; Barcena et al., 2015) and (data not shown)). We have performed preliminary assays injecting UM001 alone in nude mice but tumors grew slower than when co-injected with LX-2. Furthermore, we show that PLX51107 did not significantly alter UM001 tumor volume (Figure F) compared to tumors in mice co-injected with UM001 and LX-2.

Figure F: In vivo UM001 subcutaneous growth when injected alone or in combination with LX-2.

10) Why did you only use UM001 cells for the *in vivo* experiments? In the *in vitro* studies, other uveal melanoma cell lines were applied, as well. The quality of the study would benefit from repetition and verification of this important experiment with other cell lines.

Response: Prior to studies using UM001, we have injected another uveal melanoma cell line, UM004 *in vivo* but UM001 grew faster and into bigger tumors (data not shown). Therefore, we tested BET and FGFR inhibitors *in vivo* using UM001.

11) Besides, in the *in vivo* system there is a discrepancy during drug administration. BET inhibitors were administered every day by food uptake. However, the FGFR receptor, AZD4547, was administered by oral gavage on a "5 days on, 2 days off" scheme. Why did you not give AZD4547 every day like you did with PLX51107?

Response: We administered AZD4547 on a "5 days on, 2 days off" scheme because under this scheme, we observed weight loss of mice (Fig EV11) and hence, we did not do continuous dosing with AZD4547. However, based on a previous report on AZD4547 effects *in vivo*, we treated mice with the higher concentration of AZD4547 (Gavine et al., 2012).

12) Page 12: No difference in FGF2 levels in media conditioned by hepatocytes was described. However, data is not shown. Please provide evidence.

Response: Evidence has been added to manuscript in Fig EV10A.

13) Overall, the manuscript is written very clearly. There is just a minor comment. On page 16 there is a typo ("." before "whereas").

Response: Thank you for pointing this out. We have fixed this.

References

- Amann, T., Bataille, F., Spruss, T., Muhlbauer, M., Gabele, E., Scholmerich, J., Kiefer, P., Bosserhoff, A.K., and Hellerbrand, C. (2009). Activated hepatic stellate cells promote tumorigenicity of hepatocellular carcinoma. *Cancer Sci* 100, 646-653.
- Barcena, C., Stefanovic, M., Tutusaus, A., Martinez-Nieto, G.A., Martinez, L., Garcia-Ruiz, C., de Mingo, A., Caballeria, J., Fernandez-Checa, J.C., Mari, M., *et al.* (2015). Angiogenin secretion from hepatoma cells activates hepatic stellate cells to amplify a self-sustained cycle promoting liver cancer. *Sci Rep* 5, 7916.
- Bohm, F., Kohler, U.A., Speicher, T., and Werner, S. (2010). Regulation of liver regeneration by growth factors and cytokines. *EMBO Mol Med* 2, 294-305.
- Cronauer, M.V., Hittmair, A., Eder, I.E., Hobisch, A., Culig, Z., Ramoner, R., Zhang, J., Bartsch, G., Reissigl, A., Radmayr, C., *et al.* (1997). Basic fibroblast growth factor levels in cancer cells and in sera of patients suffering from proliferative disorders of the prostate. *Prostate* 31, 223-233.
- Dawson, M.A., Prinjha, R.K., Dittmann, A., Giotopoulos, G., Bantscheff, M., Chan, W.I., Robson, S.C., Chung, C.W., Hopf, C., Savitski, M.M., *et al.* (2011). Inhibition of BET recruitment to chromatin as an effective treatment for MLL-fusion leukaemia. *Nature* 478, 529-533.
- Filippakopoulos, P., Qi, J., Picaud, S., Shen, Y., Smith, W.B., Fedorov, O., Morse, E.M., Keates, T., Hickman, T.T., Felletar, I., *et al.* (2010). Selective inhibition of BET bromodomains. *Nature* 468, 1067-1073.
- Gavine, P.R., Mooney, L., Kilgour, E., Thomas, A.P., Al-Kadhimi, K., Beck, S., Rooney, C., Coleman, T., Baker, D., Mellor, M.J., *et al.* (2012). AZD4547: an orally bioavailable, potent, and selective inhibitor of the fibroblast growth factor receptor tyrosine kinase family. *Cancer Res* 72, 2045-2056.
- Jahagirdar, R., Attwell, S., Marusic, S., Bendele, A., Shenoy, N., McLure, K.G., Gilham, D., Norek, K., Hansen, H.C., Yu, R., *et al.* (2017). RVX-297, a BET Bromodomain Inhibitor, Has Therapeutic Effects in Preclinical Models of Acute Inflammation and Autoimmune Disease. *Mol Pharmacol* 92, 694-706.
- Jin-no, K., Tanimizu, M., Hyodo, I., Kurimoto, F., and Yamashita, T. (1997). Plasma level of basic fibroblast growth factor increases with progression of chronic liver disease. *J Gastroenterol* 32, 119-121.
- Larsson, A., Skoldenberg, E., and Ericson, H. (2002). Serum and plasma levels of FGF-2 and VEGF in healthy blood donors. *Angiogenesis* 5, 107-110.

- Luo, W., Li, J., Zhang, D., Cai, T., Song, L., Yin, X.M., Desai, D., Amin, S., Chen, J., and Huang, C. (2010). Bid mediates anti-apoptotic COX-2 induction through the IKKbeta/NFkappaB pathway due to 5-MCDE exposure. *Curr Cancer Drug Targets* 10, 96-106.
- Meng, S., Zhang, L., Tang, Y., Tu, Q., Zheng, L., Yu, L., Murray, D., Cheng, J., Kim, S.H., Zhou, X., *et al.* (2014). BET Inhibitor JQ1 Blocks Inflammation and Bone Destruction. *J Dent Res* 93, 657-662.
- Odore, E., Lokiec, F., Cvitkovic, E., Bekradda, M., Herait, P., Bourdel, F., Kahatt, C., Raffoux, E., Stathis, A., Thieblemont, C., *et al.* (2016). Phase I Population Pharmacokinetic Assessment of the Oral Bromodomain Inhibitor OTX015 in Patients with Haematologic Malignancies. *Clin Pharmacokinet* 55, 397-405.
- Ozer, H.G., El-Gamal, D., Powell, B., Hing, Z.A., Blachly, J.S., Harrington, B., Mitchell, S., Grieselhuber, N.R., Williams, K., Lai, T.H., *et al.* (2018). BRD4 Profiling Identifies Critical Chronic Lymphocytic Leukemia Oncogenic Circuits and Reveals Sensitivity to PLX51107, a Novel Structurally Distinct BET Inhibitor. *Cancer Discov.*
- Sato, N., Hattori, Y., Wenlin, D., Yamada, T., Kamata, T., Kakimoto, T., Okamoto, S., Kawamura, C., Kizaki, M., Shimada, N., *et al.* (2002). Elevated level of plasma basic fibroblast growth factor in multiple myeloma correlates with increased disease activity. *Jpn J Cancer Res* 93, 459-466.
- Wang, X., Song, Y., Jacobi, J.L., and Tuan, R.S. (2009). Inhibition of histone deacetylases antagonized FGF2 and IL-1beta effects on MMP expression in human articular chondrocytes. *Growth Factors* 27, 40-49.

3rd Editorial Decision

21 November 2018

Thank you for the submission of your revised manuscript to EMBO Molecular Medicine. We have now received the enclosed reports from the referees. As you will see the reviewers are now supportive, and I am pleased to inform you that we will be able to accept your manuscript pending minor editorial amendments. Please also take in consideration the figure changes that referee 2 is mentioning.

***** Reviewer's comments *****

Referee #1 (Comments on Novelty/Model System for Author):

This manuscript improved significantly. As a scientist in the field, now, it provides important mechanistic explanations for the results we observe in some clinical studies. Moreover, it provides in depth cues about UM-drug resistance.

Referee #1 (Remarks for Author):

We are very grateful to the authors for taking in consideration all the comments from the reviewers. The manuscript is a robust and excellent manuscript worth of publishing immediately.

Referee #2 (Comments on Novelty/Model System for Author):

systems are adequate

Referee #2 (Remarks for Author):

The current version of the manuscript is I'm proved.

Personally, I would suggest to include in the supplementary section (if possible) the data shown in the rebuttal letter as figure D since the data improve the clinical relevance and applicability fo the present study.

Along those lines, also the data shown as E in the rebuttal, might be considered for suppl. material.

Referee #3 (Remarks for Author):

The authors have revised the manuscript properly. Important in vivo experiments that were missing in the previous manuscript were performed and improved the overall quality of the study. They now demonstrate efficiency of the combination therapy of BET and FGFR inhibitors not only in a subcutaneous mouse model but also in an orthotopic liver injection model. Moreover, all previous concerns of the manuscript were answered satisfactorily. In my opinion there are no further experiments needed. Thank you for this thorough response and revision.

2nd Revision - authors' response

30 November 2018

Please take in consideration the figure changes that referee 2 is mentioning: Include in the supplementary section, figure D and E from the rebuttal letter (if possible).

Response: We have added Figures D and E from the rebuttal letter to the Appendix file. Accordingly, we have cited these figures in the manuscript text.

YOU MUST COMPLETE ALL CELLS WITH A PINK BACKGROUND

Corresponding Author Name: Andrew E. Aplin
 Journal Submitted to: EMBO Molecular Medicine
 Manuscript Number: EMM-2018-09081-V2